# A high-Arctic inner shelf–fjord system from the Last Glacial Maximum to the Present: Bessel Fjord and SW Dove Bugt, NE Greenland

Authors: Kevin Zoller[1]; Jan Sverre Laberg[1]; Tom Arne Rydningen[1], Katrine Husum[2] & Matthias Forwick[1]

[1]Department of Geosciences, UiT The Arctic University of Norway, Box 6050 Langnes, NO-9037 Tromsø, Norway [2]Norwegian Polar Institute, Box 6606 Langnes, NO-9296 Tromsø, Norway

*Correspondence to:* Kevin Zoller (kevin.zoller3@gmail.com)

## Abstract

The Greenland Ice Sheet (GrIS) responds rapidly to the present climate, therefore, its response to the predicted future warming is of concern. To learn more about the impact of future climatic warming on the ice sheet, decoding its behavior during past periods of warmer than present climate is important. However, due to the scarcity of marine studies reconstructing ice sheet conditions on the Northeast Greenland shelf and adjacent fjords, the timing of the deglaciation over marine regions and its connection to forcing factors remain poorly constrained. This includes data collected in fjords that encompass the Holocene Thermal Maximum (HTM), a period in which the climate was warmer than it is at present. This paper aims to use bathymetric data and the analysis of sediment gravity cores to enhance our understanding of ice dynamics of the GrIS in a fjord and inner shelf environment as well as give insight into the timing of deglaciation and provide a palaeoenvironmental reconstruction of southwestern Dove Bugt and Bessel Fjord since the Last Glacial Maximum (LGM). The swath bathymetry data displayed in this study is the first time the bathymetry for Bessel Fjord has become available. North-south oriented glacial lineations, and the absence of pronounced moraines in southwest Dove Bugt, an inner continental shelf embayment (trough), suggests the southwards and offshore flow of the southern branch of the Northeast Greenland Ice Stream (NEGIS), Storstrømmen. Sedimentological data suggests that an ice body, theorized to be the NEGIS, may have retreated from the region slightly before ~11.4 ka cal BP. The seabed morphology of Bessel Fjord, a fjord terminating in southern Dove Bugt, includes numerous basins, separated by thresholds. The position of basin thresholds, which include some recessional moraines, suggest that the GrIS had undergone multiple halts or readvances during deglaciation, likely during one of the cold events identified in the Greenland Summit temperature records (Kobashi et al., 2017). A minimum age of 7.1 ka cal BP is proposed for the retreat of ice through the fjord to or west of its present-day position in the Bessel Fjord catchment area. This suggests that the GrIS retreated from the marine realm in early Holocene, around the onset of the Holocene Thermal Maximum in this region, a period when the mean July temperature according to Bennike et al., (2008) was at least 2-3 ℃ higher than at present, and remained at or west of this onshore position for the remainder of the Holocene. The transition from predominantly mud to muddy sand layers in a mid-fjord core at ~4 ka cal BP may be the result of increased sediment input from nearby and growing ice caps. This shift may suggest that in the late Holocene (Meghalayan), a period characterized by a temperature drop to modern values, ice caps in Bessel Fjord fluctuated with greater sensitivity to climatic conditions than the NE sector of the GrIS.

# 1. Introduction

Ice mass loss from the Greenland Ice Sheet (GrIS) has accelerated during the 21 century, making it the largest individual contributor to sea level rise (King et al., 2020). This introduction of a substantial quantity of fresh water may have ramifications for global ocean circulations as well as the climate (Rahmstorf et al., 2015). Approximately 12% of the ice from the GrIS is transported to the coast through the Northeast Greenland Ice Stream (NEGIS) (Khan et al., 2014; Joughin et al., 2001) and therefore has a substantial impact on the mass balance of the ice sheet and a potential to contribute to sea level rise. Currently, two of the three marine terminating outlet glaciers that are supplied by the NEGIS are in retreat (Mouginot et al., 2015), where the southernmost branch, Storstrømmen in Dove Bugt (Figs. 1a & 1b), is currently in a building phase following a 1978-1984 surge (Khan et al., 2014; Reeh et al., 1994). While there are numerous modern studies on the current state of the NEGIS during the past decades to century, there is a scarcity of data concerning the position and dynamics of the ice stream, and other local Northeast Greenland outlet glaciers, on a multi-century to millennia scale over marine regions. Considering that the global mean temperature is expected to continue to rise (Stocker et al., 2013), and that the Arctic will experience an amplification effect (Cohen et al., 2014), looking to the past, especially during warmer than present periods (i.e., the Holocene Thermal Maximum (HTM)), may provide an important insight into the future behavior of the ice sheet.

Marine studies have found evidence for past advancement and retreat of the GrIS and NEGIS along the continental shelf offshore Northeast Greenland (Evans et al., 2009; Winkelmann et al., 2010; Arndt et al., 2015, 2017; Laberg et al., 2017; Arndt, 2018; Olsen et al., 2020; Syring et al., 2020; Davies et al., 2022; Hansen et al., 2022; Jackson et al., 2022). Geomorphological findings in Store Koldewey Trough (~76°N), a major shelf trough northeast of the study area (Fig. 1b), suggests that the ice sheet may have reached the shelf break in this area during the LGM (Last Glacial Maximum) (Laberg et al., 2017; Olsen et al., 2020). Further north (~79.4°N), the shelf break is interpreted as being ice free during the LGM (Rasmussen et al., 2022), an area where the ice front had its maximum LGM position at the outer shelf according to Arndt et al. (2017). A concise understanding of the timing and dynamics of the ice sheet over the NE Greenland shelf during the subsequent deglaciation of the marine realm remains to be established as very few dated cores have been recovered. Terrestrial dating (e.g., cosmogenic nuclide dates and lake studies) has provided further insight into when terrestrial regions had become deglaciated, and how the climate has changed in these areas (e.g., Björck and Persson, 1981; Björck et al., 1994; Wagner et al., 2008; Klug et al., 2009a; Schmidt et al., 2011; Briner et al., 2016; Skov et al., 2020; Larsen et al., 2020). However, only recently has terrestrial data been integrated with marine data to establish a detailed deglaciation chronology of the shelf, coastal and fjord regions (Davies et al., 2022; Larsen et al., 2022).

Swath bathymetry and gravity cores data from southwestern Dove Bugt (i.e., Store Bælt) and Bessel Fjord (Fig. 1), presented for the first time in this study, has been used to further refine our understanding of how the GrIS responded to changes in palaeoclimatic conditions from the LGM through the Holocene, including the HTM. Through this analysis we aim to reconstruct regional ice dynamics from both full-glacial conditions and during overall retreat and put our findings into the larger context of the dynamics of the Northeast Greenland Ice Sheet during these periods. Additionally, this study aims to refine our understanding about the timing of deglaciation over marine areas and compare findings to nearby terrestrial regions including the

Store Koldewey island and Hochstetter Forland/Shannon Ø. Results will also contribute to our
understanding of palaeoenvironmental conditions throughout the Holocene for the NE
Greenland fjords and inner shelf areas.

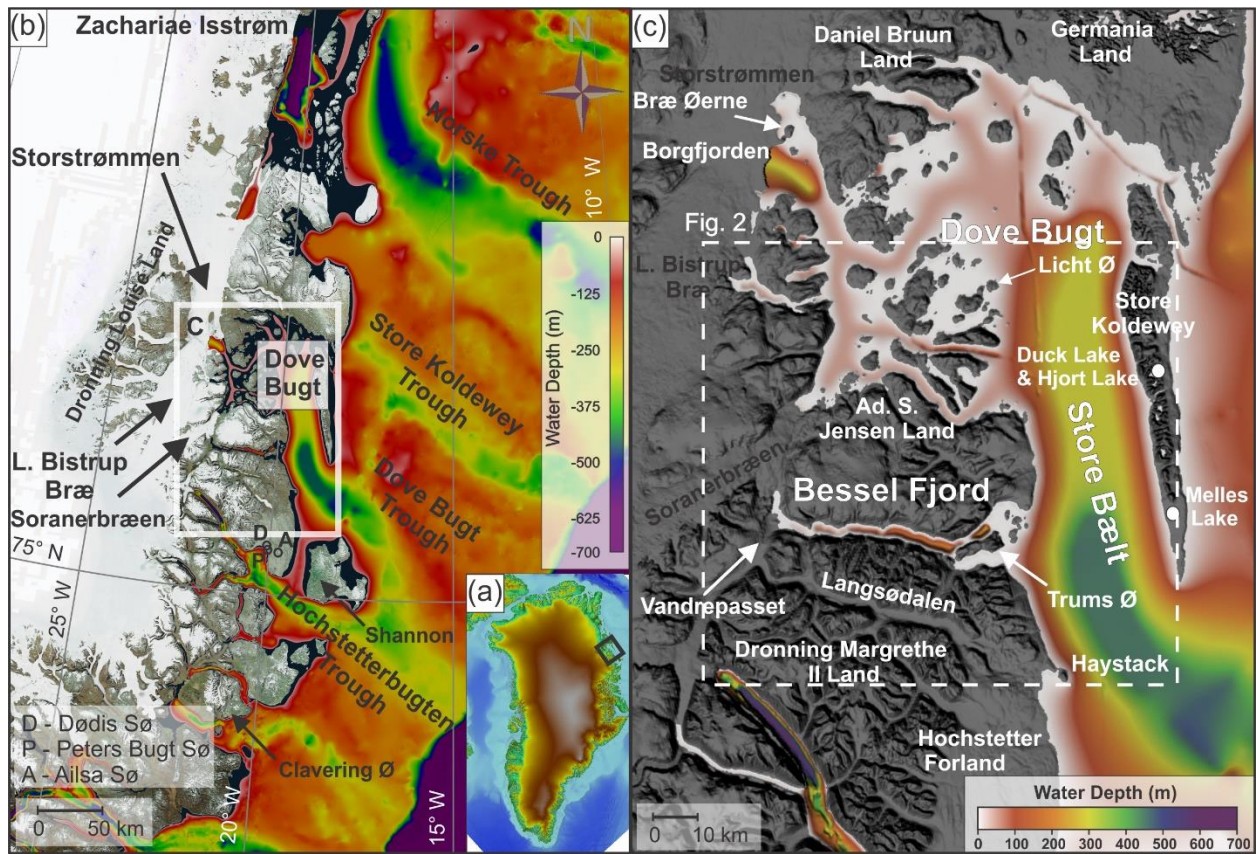


*Figure 1. (a) An image of Greenland, using IBCAO 4.0 400x400m (Jakobsson et al., 2020), with a black box*
*surrounding the study area. (b) Bathymetry of Northeast Greenland displayed using IBCAO 4.0 200x200m data*
*(Jakobsson et al., 2020) and land is displayed using a World Imagery satellite image (Earthstar Geographics, Esri,*
*HERE, Garmin, FAO, NOAA, USGS) made available through GlobalMapper. The white box surrounds the position of*
*Fig. 1c. (c) Bathymetry of Dove Bugt and Bessel Fjord and surrounding land areas displayed using the IBCAO 4.0*
*200x200m data (Jakobsson et al., 2020). Locations mentioned in the text are labelled here. The position of Fig. 2 is*
*within the white dashed box.*

## 2. Regional Setting and Environmental History

Bessel Fjord is a west-east running fjord between Adolf S. Jensen Land and Dronning
Margrethe II Land (Fig. 1c). The western end of the fjord contains the southern outlet glacier
Soranerbræen, which also has a second outlet to the north in a tributary fjord to inner Dove Bugt
(Fig. 2). Several ice caps are positioned across the length of the fjord (Figs. 2 & 3), some of
which have several generations of moraines and glaciofluvial outlets that enter the fjord.
Colluvial fans and rivers have been observed across the length of the fjord in satellite images
and while surveying the fjord. Multiple islands are located at the entrance of Bessel Fjord, the
largest of which, Trums Ø, splits the entrance into two main inlets (Figs. 1c & 2). From the
termination of Soranerbræen to the entrance of the fjord measures ~60 km in length. The width
of the fjord ranges from 1.8 to 3.7 km.
To the west of Bessel Fjord and Soranerbræen is the larger glacier L. Bistrup Bræ, which flows
northwards and has an outlet in Borgfjorden, another tributary fjord to inner Dove Bugt (Fig 1).
Here it is confluent with the southward flowing NEGIS outlet glacier, Storstrømmen (Rignot et
al., 2022). Studies of modern Soranerbræen, L. Bistrup Bræ and Storstrømmen suggest that
they all have separate drainage basins (Krieger et al., 2020). Storstrømmen and L. Bistrup Bræ
are two of the largest surge-type glaciers in the world (Higgins, 1991) with a surge periodicity of
approximately 70 years (Mouginot et al., 2018).
Bathymetry of inner Dove Bugt and tributary fjords has revealed that there are no natural large
passageways for the warm, salty, subsurface Atlantic Intermediate Water to impact these
glaciers at present, therefore it has been suggested that ocean waters do not play a large role in
the evolution of Storstrømmen, L. Bistrup Bræ and the northern outlet of Soranerbræen, and
that their grounding line retreat is mostly caused by ice thinning (Rignot et al., 2022).
Mega-scale glacial lineations (MSGL) identified in Store Koldewey Trough on the continental
shelf have been interpreted as evidence for the expanse of this sector of the GrIS to the shelf
break during the LGM (Laberg et al., 2017; Olsen et al., 2020). This is further supported by the
presence of recessional moraines and grounding zone wedges, which suggests a complex
deglaciation of this part of the shelf area (Arndt et al., 2015, 2017; Laberg et al., 2017; Arndt,
2018; Olsen et al., 2020). Olsen et al. (2020) has suggested that deglaciation in the Store
Koldewey Trough may have occurred in two stages: first, an initial retreat as a result of eustatic
sea level rise caused by melting ice at lower latitudes (Lambeck et al., 2014), followed by a
melting phase driven by ocean warming. So far, the timing of the onset of the deglaciation is not
known. Across the GrIS, deglaciation is believed to be asynchronous, with factors such as
topography and local ice dynamics playing a large role with ice retreat in conjunction with
climate change (Bennike & Björck, 2002; Funder et al., 2011; Ó Cofaigh et al., 2013; Hogan et
al., 2016).
A recent study by Jackson et al. (2022) of the inner shelf east of the Clavering Ø (~74°N; Fig.
1b) indicated that during the late Younger Dryas, this sector of the GrIS had reached a more
landward position, in conformity with Funder et al. (2021). During this period the inner shelf
bottom water was characterized by anomalously high temperatures, interpreted to have played
a role in the ice retreat and leading to the termination of the Younger Dryas stadial. This was
followed by the onset of the East Greenland Current, as seen from cooler bottom water from the
Early Holocene on (Jackson et al., 2022).
Further north, east of marine terminating glacier Zachariae Isstrøm (~78° 30N; Fig. 1b), the
deglaciation of the NEGIS from the inner shelf was found to have occurred as early as 12.5 ka
cal BP, likely before 13.4 ka cal BP. Here, inflow of warmer water (Atlantic Water) may have
played a role. This part of the shelf was covered by an ice shelf from 13.4 to 11.2 ka cal BP
(including the Younger Dryas), retreating and leading to open water conditions from the earliest
Holocene; 11.2-10.8 ka cal BP, before readvancing from 10.8 to 9.6 ka cal BP, finally retreating
from 9.6 to 7.9 ka cal BP. At 7.9 ka cal BP there was a drastic shift in ocean circulation at this
site with a sharp decline in Atlantic Water corresponding to an increase in Polar Water influx
(Davies et al., 2022). Pados-Dibattista et al. (2022), studying another core from the NE
Greenland shelf (more seaward, in a mid-shelf position north of the Norske Trough at ~79°N),
found that during the early Holocene (9.4 to 8.2 ka cal BP), the East Greenland Current was
highly stratified with cold surface water overlying warm Atlantic subsurface water.

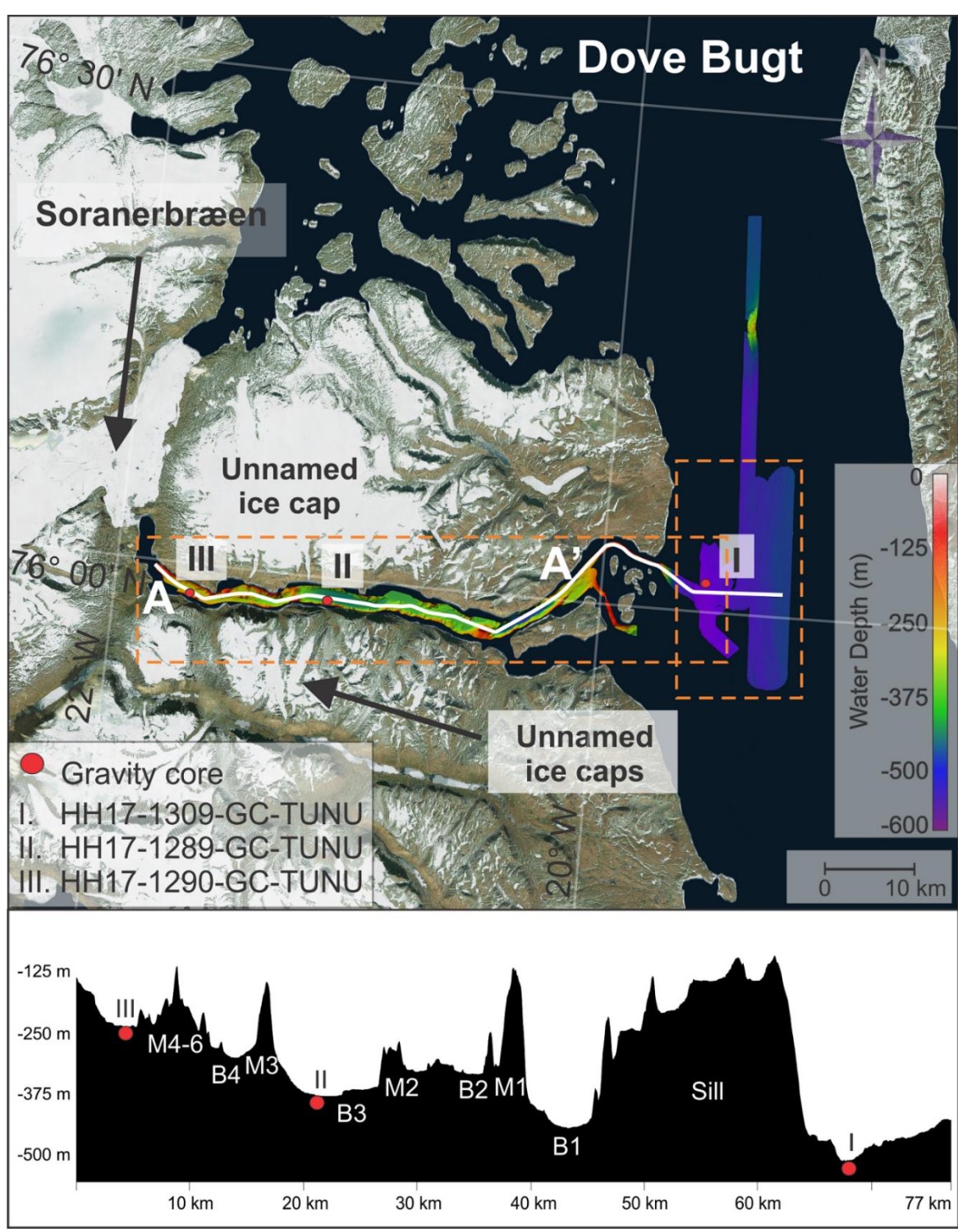

*Figure 2. Study area with the bathymetric data showing the locations of the sediment cores presented in this study. The lower panel is a profile along the length of Bessel Fjord, A-A'. Sediment cores are labelled I, II and III. Satellite image is displayed using a World Imagery satellite image (Earthstar Geographics, Esri, HERE, Garmin, FAO, NOAA, USGS) made available through GlobalMapper.*

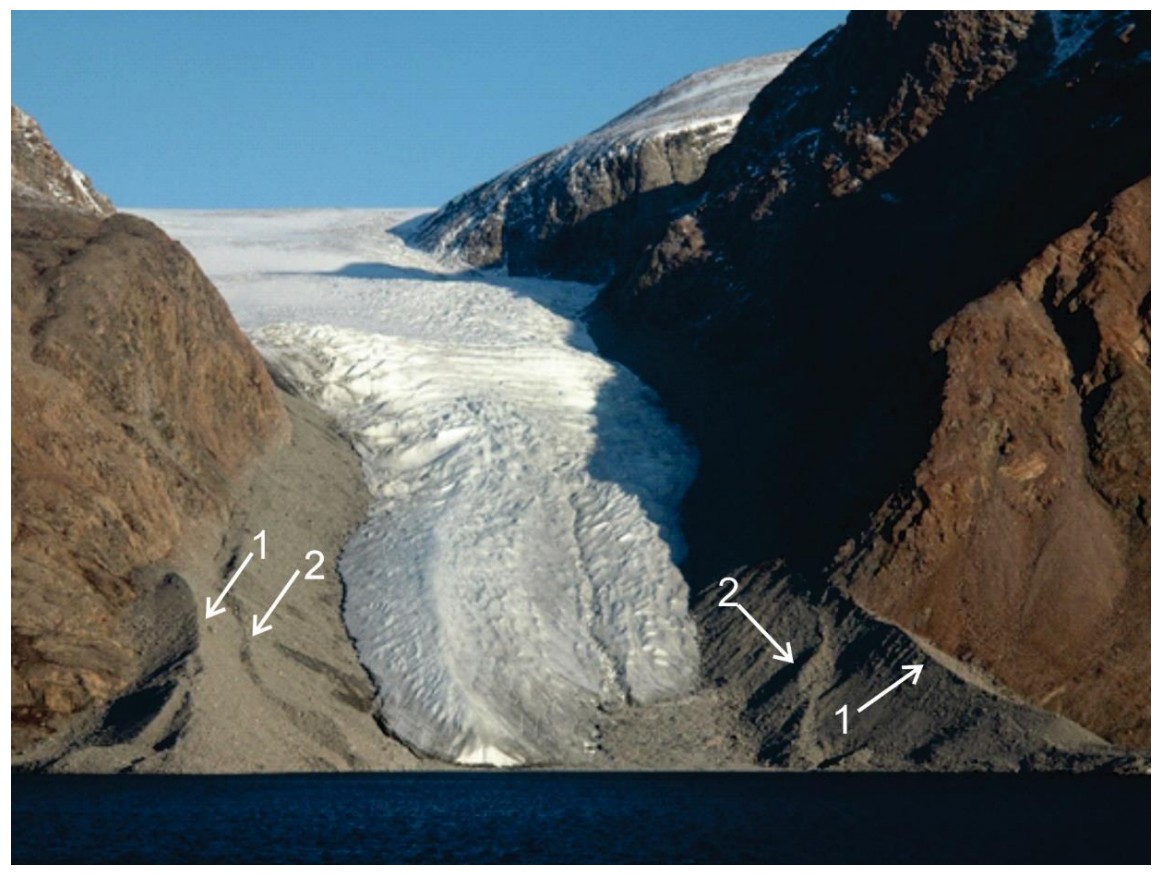

*Figure 3. Image of an ice lobe from an ice cap near gravity core HH17-1289-GC-TUNU. Two sets of coarse-grained terminal morainal ridges are indicated by numbers and arrow. See Fig. 6b for the position of the modern ice lobe. The photograph was taken by Torger Grytå on a 2017 TUNU cruise.*

Following the 8.2 ka event, the interval from 8.2 to 6.2 ka cal BP was followed by the warmest Holocene bottom water conditions on the shelf. Afterwards, conditions returned to those seen prior to 8.2 ka cal BP due to increased Polar Water transport strengthening the East Greenland Current.

Terrestrial studies of Dronnings Margrethe II Land, Germania Land and adjacent areas have identified a complex assortment of moraines that are believed to have formed during the Kap Mackenzie, Muschelbjerg, Nanok I and Nanok II stadials (Hjort, 1979, 1981; Hjort and Björck, 1983; Björck et al., 1994; Landvik, 1994). The exact ages of these stadials remain unclear (Table 1), yet Larsen et al. (2022) suggests that Nanok-stadial moraines found in Store Koldewey formed synchronously with the Milne Land moraines of Scoresby Sund which date to the Allerød to early Younger Dryas and Preboreal time (Kelly et al., 2008; Levy et al., 2016).

The position of striations on Store Koldewey and lateral moraines on coastal slopes between Bessel Fjord and Haystack have been interpreted as evidence for ice flowing out of Dove Bugt and Bessel Fjord during the Muschelbjerg stadial, southwards through Store Bælt and turning eastwards around the southernmost mountains of Store Koldewey (Hjort, 1981). Early studies of the region noted glacial and glaciofluvial deposits (e.g., moraine plateaux, terminal moraines, eskers and sandurs) on Hochstetter Forland that are believed to have formed during this period (Hjort, 1979, 1981).

*Table 1. Previously published stadial information for the Dove Bugt region as well as age estimates used in this study.*

| Stadials | Studies | | | | | Age estimate used in this study |
|---|---|---|---|---|---|---|
| | *Hjort & Björck (1983)* | *Funder et al., (1998)* | *Kelly et al. (2008)* | *Vasskog et al. (2015)* | *Larsen et al. (2022)* | |
| *Nanok II* | 10.1-9.5 ka cal BP | Preboreal (ending at ca. 9.7 ka cal BP) | Younger Dryas and Early Holocene (13-11.6 ka cal BP (G-III), 11.7-10.6 ka cal BP (G II)) | Close to Bølling– Allerød transition, and late Younger Dryas (~14 ka cal BP (G III), ~12 ka cal BP (G-II)) | Preboreal | Preboreal |
| *Nanok I* | Older than 14 ka cal BP, possibly between 15 and 19 ka cal BP | | | | Late Allerød to early Younger Dryas | Late Allerød to early Younger Dryas |
| *Nanok 0* | | ~48 ka (Hjort, unpublished data) | | | | ? |
| *Muschelbjerg* | Saalian (or older)? | | | | | Saalian (or older)? |
| *Kap Mackenzie* | Saalian (or older)? | | | | | Saalian (or older)? |

Lateral moraines and glacial striations oriented along the axis of Langsodal (also referred to as Langsødalen; Fig. 1c), a nearby valley south of and sub-parallel to Bessel Fjord, have been interpreted as evidence for glacial confinement within the valley during an undifferentiated Nanok stadial (Hjort 1979; Hjort, 1981). This differs from striations that have also been identified in the valley along more weathered surfaces that are oriented in a southwestern direction (Hjort, 1979).

The outer coastal regions of North and Northeast Greenland are believed to have been deglaciated between 12.8 and 9.7 ka cal BP and present ice positions were reached between 10.8 to 5.8 ka cal BP (Larsen et al., 2022). Cosmogenic nuclide dates from Store Koldewey, first collected by Håkansson et al. (2007), and later Skov et al. (2020) and Larsen et al. (2022), suggest that ice retreated from the continental shelf and reached the upper and lower sections of the island by 12.3 and 12.7 ka cal BP, respectively. In contrast, Biette et al. (2020) found evidence of the deglaciation of Clavering Ø at 16.2 ka cal BP, with readvances at 11.3, 10.8, 3.3, 1.2 and 0.37 ka cal BP. Additional cosmogenic nuclide findings indicate that Trums Ø, in

outer Bessel Fjord, may have become deglaciated around 12.6 ka cal BP and Vandrepasset,
onshore inner Bessel Fjord, by 8.6 ka cal BP (Larsen et al., 2022).
Findings from macrofossil remains (Bennike & Björck, 2002) and lacustrine sedimentary records
(Cremer et al., 2008) suggest that coastal regions were deglaciated in a ~1500 year span after
the start of the Holocene (Klug et al., 2016).  To the north of Store Koldewey, a minimum date
for deglaciation in Germania Land of 9.5 ka cal BP has been proposed (Landvik, 1994),
whereas to the south in southern Dronning Margrethe II Land, a minimum date of 11.2 ka cal BP
has been suggested (Bennike & Weidick, 2001). Lake studies on aquatic organisms at Björck
Lake and Hjort Lake on Store Koldewey (Fig. 1c) indicate that the island was at its warmest
between ~8 and 4 ka cal BP, (Wagner et al., 2008; Klug et al., 2009; Schmidt et al., 2011),
although findings from Melles Lake (Fig. 1c) suggest that the earliest onset of warmth during the
Holocene may have occurred at ~ 10 ka cal BP (Klug et al., 2009; Briner et al., 2016). On
Hochstetter Forland (Fig. 1c), pollen assemblages from Dødis Sø, Peters Bugt Sø and Ailsa Sø
suggest that the temperatures were at their highest between 8.8 and 5.6 ka cal BP (Björck &
Persson, 1981; Björck et al., 1994). These findings indicate that the HTM was not uniform
across East Greenland, as also described by Briner et al. (2016).
To the south, offshore the Kejser Franz Josef fjord system (~73°N), a detailed biomarker record
finds this part of the shelf dominated by seasonal sea ice throughout the late Holocene (<~5 ka
cal BP) and extended concentrations from 5.2 to 2.2 and 1.3 to present. Short-term variability
was also seen for this area for the last 2.2 ka cal BP, corresponding to the climatic events of this
period (Kolling et al., 2017).

## 222 3. Material and Methods
Swath bathymetry and three sediment cores were collected in southwestern Dove Bugt and
Bessel Fjord during an expedition aboard RV *Helmer Hanssen* of UiT The Arctic University of
Norway in September 2017, being part of the TUNU program (Fig. 2; Christiansen, 2012). The
swath bathymetry data was obtained using a Kongsberg Maritime Simrad EM 302 multibeam
echo sounder. It was gridded using Petrel software, and geomorphological interpretations were
made using Global Mapper 18. Surfaces were developed using a 5x5m grid cell size while a
surface created from an International Bathymetric Chart of the Arctic Ocean (IBCAO) dataset
4.0 with a 200x200m grid cell size (Jakobsson et al., 2020).
Two soft sediment gravity cores were retrieved from Bessel Fjord (HH17-1289-GC-TUNU &
HH17-1290-GC-TUNU) and one southwest of Dove Bugt in the sound Store Bælt (HH17-1309-
GC-TUNU) (Fig. 2 & Table 2). Prior to splitting the cores, physical properties were measured
using a GEOTEK Multi Sensor Core Logger (MSCL-S). The cores were placed in the laboratory
for 24 hours prior to obtaining physical measurements to ensure that each core temperature
reached equilibrium with the laboratory to avoid distorting p-wave values (Weber et al., 1997).
A GEOTEK MSCL X-ray Computed Tomographic imaging machine was also used to scan the
unopened core sections to create X-ray radiographic images. After each core was split and
cleaned, the characteristics of the sedimentary surface were logged (i.e., structures,
bioturbation, grain size, lithological boundaries, etc.), sediment color was noted using the
Munsell Soil Color Chart and lithofacies were assigned based on Eyles et al. (1983)
classification system. X-ray fluorescence (XRF) data (not published here), as well as colored
images of the core sections, were then obtained using an Avaatech XRF core scanner.

 *Table 2. Information on the position, water depth and recovery length of each gravity core. Note that the core names*
*are abbreviated in the text.*

| Location | Inner Bessel Fjord | Mid-Bessel Fjord | Southeastern Dove Bugt |
|---|---|---|---|
| Coring station | HH17-1290 | HH17-1289 | HH17-1309 |
| Latitude [N] | 75° 58' 34.5907" | 75° 58' 11.4928" | 76° 01' 34.0387" |
| Longitude [W] | 21° 07' 13.1055" | 21° 41' 48.0278" | 19° 34' 31.3190" |
| Water depth [m] | 372 | 225 | 512 |
| Recovery [cm] | 534.5 | 245.5 | 474.55 |


Molluscs and benthic foraminifera were recovered from each core for the purpose of
radiocarbon dating of lithofacies boundaries. This was, however, not always possible due to the
low content of foraminifera and molluscs in these cores which also restricted the number of
dates that could be obtained. Two adjacent 1 cm thick sediment slices were successfully
sampled from select positions across cores HH17-1290 and HH17-1309. Samples were then
wet sieved at 1 mm, 100 μm and 63 μm meshes, respectively. Benthic foraminifera from the
100-um size fraction were extracted for radiocarbon dating. Radiocarbon dating was carried out
at the MICADAS radiocarbon laboratory at Alfred Wegener Institute, Helmholtz Centre for Polar
and Marine Research, Germany. The radiocarbon dates were calibrated using the online
version of OxCal 4.4 (https://c14.arch.ox.ac.uk/oxcal.html#program) and the Marine20
calibration curve (Heaton et al., 2020), as the calibrated 14C samples are younger than 11.5 ka
cal BP (Heaton et al., 2022). We are using a ΔR of -10 ± 60 in conformity with Jackson et al.
(2022). Previously reported radiocarbon dates from this area that are relevant to our study have
been recalibrated using Marine20 for marine samples under 11.5 ka and IntCal20 for terrestrial
samples (Reimer et al., 2020). One marine sample older than 11.5 ka cal BP has also been
included (Table 3). We are aware that for the Arctic, including our study area, calibration of
marine samples by Marine20 is not recommended for samples older than 11.5 cal ka BP (see
Heaton et al. (2022)), therefore, this calibrated age is treated with caution.
A Beckman Coulter LS 13 320 Multi-Wavelength Laser Diffraction Particle Size Analyzer was
used to perform sediment grain size analysis. Sediment was sampled in mostly 10 cm intervals
across HH17-1309, where samples taken from the other two cores were selected from specific
positions. Samples were treated in HCl and $H_2O_2$ and a pre-heated VWB 18 Thermal Bath.
Samples were then cleaned using distilled water, placed through multiple runs through a
centrifuge and heated in an oven to remove water content. Approximately 0.2 grams of
sediment were then separated and placed in a container with 20 ml of water and moved to a
shaking table for over 48 hours. A few drops of Calgon were added to each sample, which was
then placed into a Branson 200 ultrasonic cleaner for ~7 minutes and shaken briefly before
being poured through a >2 mm mesh and into the particle size analyzer. Grains between the
size of 0.4 μm and 2000 μm were counted and underwent three separate runs. GRADISTAT
Excel-software was used to calculate the mean of the three runs. Sediment names used in
reference to this analysis are based on Folk (1954) and mean grain size from the methodology
published by Folk & Ward (1957).


*Table 3. Other published radiocarbon dates and their recalibrated ages using Marine20 (and an ΔR of -10 ± 60 in*
*conformity with Jackson et al. (2022)) and IntCal20 for aquatic moss samples. *The age of sample Lu-1298 from*
*Shannon is above what is recommended by Heaton et al., (2022) for use with Marine20 and is therefore treated with*
*caution.*

| Location | Material | Lab nr. | $^{14}$C age | $^{14}$C cal BP (1 σ range) | $^{14}$C cal BP (median) | Reference |
|---|---|---|---|---|---|---|
| Shannon | shell | Lu-1298* | 19000 ± 190 | 21855-22325 | 22078 | Hjort, 1981; Hjort 1979 |
| Hochstetter F. | shell | Lu-1289 | 9190 ± 90 | 9572-9926 | 9779 | Hjort, 1981; Hjort 1979 |
| Shannon | shell | Lu-1389 | 9370 ± 90 | 9865-10195 | 10015 | Hjort, 1981; Hjort 1979 |
| Hochstetter F. | shell | Lu-1386 | 9400 ± 90 | 9896-10220 | 10054 | Hjort, 1981; Hjort 1979 |
| Hochstetter F. | shell | Lu-1300:1 | 9470 ± 90 | 9970-10322 | 10157 | Hjort, 1981; Hjort 1979 |
| Hochstetter F. | shell | Lu-1300:2 | 9520 ± 90 | 10084-10412 | 10229 | Hjort, 1981; Hjort 1979 |
| Hochstetter F. | shell | Lu-1384 | 9810 ± 95 | 10409-10794 | 10617 | Hjort, 1981; Hjort 1979 |
| Ardencaple Fjord | shell | Lu-1390 | 8570 ± 85 | 8864-9200 | 9022 | Hjort, 1981; Hjort 1979 |
| Kildedalen | shell | Lu-1303 | 8930 ± 90 | 9290-9573 | 9447 | Hjort, 1981; Hjort 1979 |
| Snenæs | Mya truncata, Hiatella arctica | T-9372 | 8265 ± 95 | 8434-8768 | 8619 | Landvik 1994 |
| Hvalrosodden moraine | Nuculana pernula | TUa-123 | 8685 ± 95 | 9006-9315 | 9166 | Landvik 1994 |
| Hvalrosodden moraine | Nuculana pernula | TUa-124 | 9045 ± 90 | 9438-9741 | 9596 | Landvik 1994 |
| Hvalrosodden | Mya truncata | T-9361 | 8190 ± 95 | 8360-8663 | 8523 | Landvik 1994 |
| Hvalrosodden | Mya truncata, Hiatella arctica | T-9370 | 7930 ± 120 | 8681-9085 | 8890 | Landvik 1994 |
| Hvalrosodden | Mya truncata | T-9371 | 7490 ± 115 | 8186-8502 | 8348 | Landvik 1994 |
| Peters Bugt | Portlandia arctica | Ua-2787 | 10260 ± 105 | 11071-11444 | 11253 | Björck, 1994 |
| Peters Bugt Sø | Hiatella arctica | Lu-3516 | 9640 ± 90 | 10222-10527 | 10382 | Björck, 1994 |
| Storstrømmen Sound | Mya truncata & Hiatella arctica | K-6098 | 5180 ± 95 | 5220-5520 | 5352 | Weidick et al., 1994 |
| Storstrømmen Sound | Mya truncata | K-5494 | 4910 ± 85 | 4865-5175 | 5028 | Weidick et al., 1994 |
| Storstrømmen Sound | Mya truncata | K-5493 | 4840 ± 90 | 4793-5117 | 4943 | Weidick et al., 1994 |
| Storstrømmen Sound | Hiatella arctica | Ua-3347 | 5030 ± 75 | 5023-5311 | 5166 | Weidick et al., 1994 |
| Storstrømmen Sound | Hiatella arctica | Ua-3350 | 4180 ± 60 | 3944-4225 | 4082 | Weidick et al., 1994 |
| Storstrømmen Sound | Balanoptera physalus | K-6096 | 3630 ± 90 | 3230-3530 | 3380 | Weidick et al., 1994 |
| Storstrømmen Sound | Hiatella arctica | Ua_3349 | 3725 ± 60 | 3371-3616 | 3496 | Weidick et al., 1994 |
| Storstrømmen Sound | Hiatella arctica & Mya truncata | K-6097 | 3230 ± 85 | 2749-3024 | 2897 | Weidick et al., 1994 |
| Storstrømmen Sound | Hiatella arctica | Ua-3348 | 1815 ± 55 | 1115-1317 | 1217 | Weidick et al., 1994 |
| Hjort Lake | Warnstorfia exannulata | Poz-6194 | 8260 ± 50 | 8456-8722 | 8602 | Wagner, 2008 |
| Duck Lake | Aquatic moss | LuS-6525 | 8690 ± 230 | 9527-10145 | 9775 | Klug 2009 |






# 4. Results

## 4.1. Seafloor landforms in SW Dove Bugt (Store Bælt)

### 4.1.1. Elongated Lineations - Glacial Lineations

Slightly curved sub-parallel lineations, oriented sub-parallel to the axis of Dove Bugt, are the most pronounced landforms in this part of the study area. They are oriented N-NW in the south and N-NE in the north (Fig. 4). The most frequently identified positive lineations (ridges) are 35-50 m in width, <1-3 m in height and between 1 and 10 km in length. Length to width ratios are frequently >10:1. At elevations shallower than 435 m depth, near the center of Store Bælt, the lineations are wider (e.g., 60-150 m wide), and occasional merging and overlapping of lineations occur (Fig. 4e). Wider lineations, often identified in the southern section of the study area (Fig. 4b), have also been identified with widths, lengths and heights ranging from 200-650 m, 3-8 km and 4.5-15 m, respectively. Length to width ratios here are 7:1 to >10:1. Some of the larger lineations are superimposed by smaller lineations. Lateral ridges have also been identified in clusters overprinting the lineations (Fig. 4c), where furrows have been found cross cutting lineations (Fig. 4d). Lateral ridges measure 0.5 to 2 m in height and are approximately 45 to 250 m apart.

These elongated lineations are interpreted as glacial lineations (e.g., Ó Cofaigh, 2005). The thinner, more common lineations (with length/width-ratios >10:1) have been interpreted as mega-scale glacial lineations (MSGL), and such landforms are commonly associated with palaeo-ice stream environments (e.g., Stokes & Clark, 2001). Glacial lineations have been identified in numerous continental shelf regions around Greenland (Evans et al., 2009; Dowdeswell et al., 2014; Slabon et al., 2016; Laberg et al., 2017; Newton et al., 2017; Arndt, 2018; Batchelor et al., 2018; Jakobsson et al., 2018). While the mechanism behind the formation of these features are still being debated, some authors have suggested that they may have formed through meltwater flooding (Shaw et al., 2008), groove-ploughing (Clark et al., 2003) or the transverse flow in basal ice (Schoof and Clarke, 2008). King et al. (2009), who viewed the formation of MSGL in real time in West Antarctica favored aspects of the dilatant till instability model, but with till properties that could explain ribbed moraine formation and the development of these landforms on a decadal timescale. Sets of ridges that overprint the glacial lineations have been interpreted as recessional moraines, where furrows have been interpreted as iceberg plough marks.

### 4.1.2. Depression and Mound- Hill-Hole Pair

In northern Store Bælt, a 200 by 450 m wide, 3-4 m deep depression has been identified next to a mound with a width and height of 235 by 450 m and 3-4 m, respectively (Fig. 4d). The depression overprints N-S trending lineations, although the mound contains lineations on its surface.

This depression and mound have been interpreted as a hill-hole pair. These landforms can form when ice-thrust rafts of sediment are removed from the bed by cold-based, slow-flowing ice that transports the sediment that was once in the depression (Hogan et al., 2010; Klages et al., 2013, 2015). In this instance, a south bound ice stream may have removed frozen sedimentary material and deposited it further south.

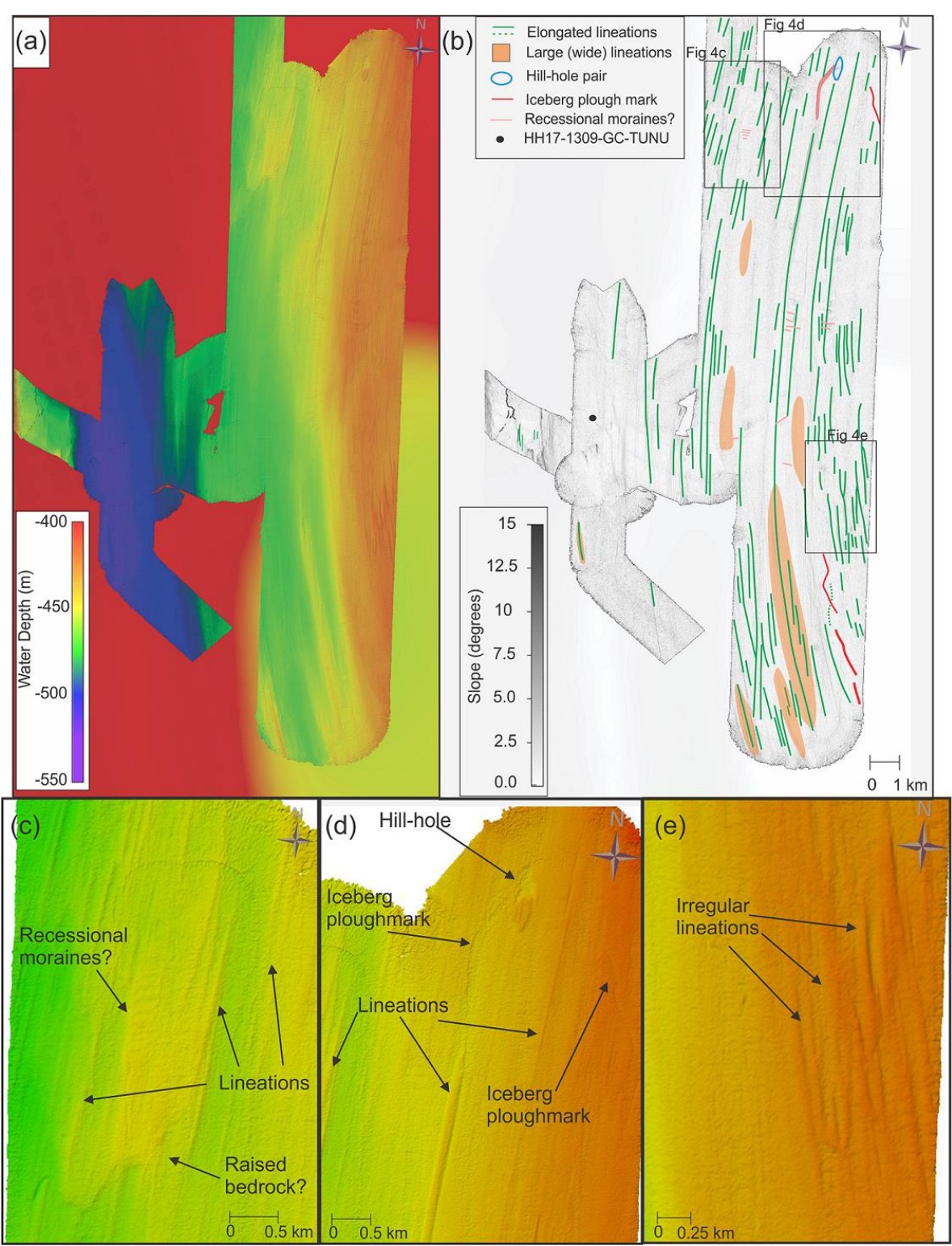

332

*Figure 4. Bathymetric maps from SW Dove Bugt. (a) Seafloor relative to water depth with IBCAO 4.0 displayed in the*
*background (Jakobsson et al., 2020). (b) The main landforms and slope angles of the seafloor in SW Dove Bugt.*
*Locations of Figs. 4c-e are indicated. (c) Bathymetry of the northwestern section of the study area. (d) Bathymetry of*
*the northeaster part of the study area. (e) Bathymetry of the eastern part of the study area showing irregularly shaped*
*glacial lineations.*

## 4.2.	Sea floor landforms in Bessel Fjord

### 4.2.1. Large scale geomorphology

Bessel Fjord contains a variety of basins that are separated by different styles of sills (Figs. 2, 5 & 6). The outermost sill is at the fjord's entrance, and it commonly ranges in depth from 50 to 200 m, with major sections reaching above (and near) the water surface as there are islands in the fjord entrance. Four large basins that are elongated in a west-east direction have been identified in Bessel Fjord (B1-B4). The deepest basin, Basin 1 (B1), is the closest to the fjord entrance and is separated from basin 2 (B2) by a >215 m high sill (M1) that is steeper to the east (Figs. 2 & 5). Basin 3 progressively deepens westwards, with a maximum depth of 380 m. A ~70 to 160 m asymmetrical sill (M3; Figs. 2 & 5) that is steeper on its east side separates Basin 3 from basin 4. Basin 4 is the shallowest basin (~280-300 m) and is adjacent to multiple smaller basins that are primarily at lower points of elevation. The fjord also contains smaller basins that are raised relative to the average seafloor depth (Fig. 6e). Features interpreted as bedrock mounds have also been identified in other sections of the fjord (Figs. 5 & 6). Along the fjord sides, landforms from sediment reworking including slide scars, channels and gullies have also been observed Fig 6b.

### 4.2.2. Linear Ridges Oriented Along Fjord Axis- Glacial Lineations

Oriented along the fjord's axis (or at times slightly oblique to it), linear features have been identified in the inner and middle of the fjord, as well as a single lineation on the outer part of the fjord (Figs. 5 & 6). They range in size from 100 to 1000 m in length and ~3 to 9 m in height, although some that are as high as 80 m have been identified in the inner fjord. Their morphologies vary throughout the fjord, and their length to width ratios range from 2:1 to 5:1. Most ridges slope towards the outer fjord, although some slope in the opposite direction or have an irregular or flat top. They appear both independently in connection with inferred bedrock highs, and in clusters in flat lying areas of basin 3. These ridges have been interpreted as glacial lineations, and they are thus indicating the direction of former glacier flow.

### 4.2.3. Transverse Ridges- Moraines

Several transverse ridges have been identified in the inner and central portion of the fjord, oriented perpendicular to the fjord's axis (Figs. 2, 5 & 6). The ridges in the inner most position of the fjord tend to largely conform to the topography (i.e., between bedrock mounds, some of which are position mid-fjord (M4-6; Fig. 6b), and the fjord sidewalls) and are the threshold between sub-basins (Fig. 6). The width and length of ridges range from 150 to 600 m and 120 to 500 m, respectively, where their heights are between <5 to 58 m.

A particularly large, asymmetrical transverse ridge that spans the width of the fjord, is situated between Basin 3 and 4 (M3; Figs. 2 & 6d). This ridge is ~1.5 km in width and between 72 to 162 m in height. It contains a crescent shape in aerial view and is concave towards the mouth of the fjord. A large threshold with a 1.8 km width and a > 215 m height also separates basin 1 and 2 (M1; Figs. 2 & 5). This feature is ~150m shallower in the north and dips steeply into basin 1.

The transverse ridges have been interpreted as moraines, which would have formed during glacial stillstands or readvancements during the retreat of a grounded tidewater glaciers margin. These moraines do not fill the width of the innermost fjord, which has also been seen in inner Nordfjord (part of the Keiser Franz Josef fjord system) by Olsen et al. (2022). While the large transverse ridge M3 is believed to be a moraine, it is considered more likely that M1 is a bedrock mound based on its morphology. The smaller transverse ridges are interpreted as recessional moraines. Smaller moraines have the potential to form at ice margins annually

383

384

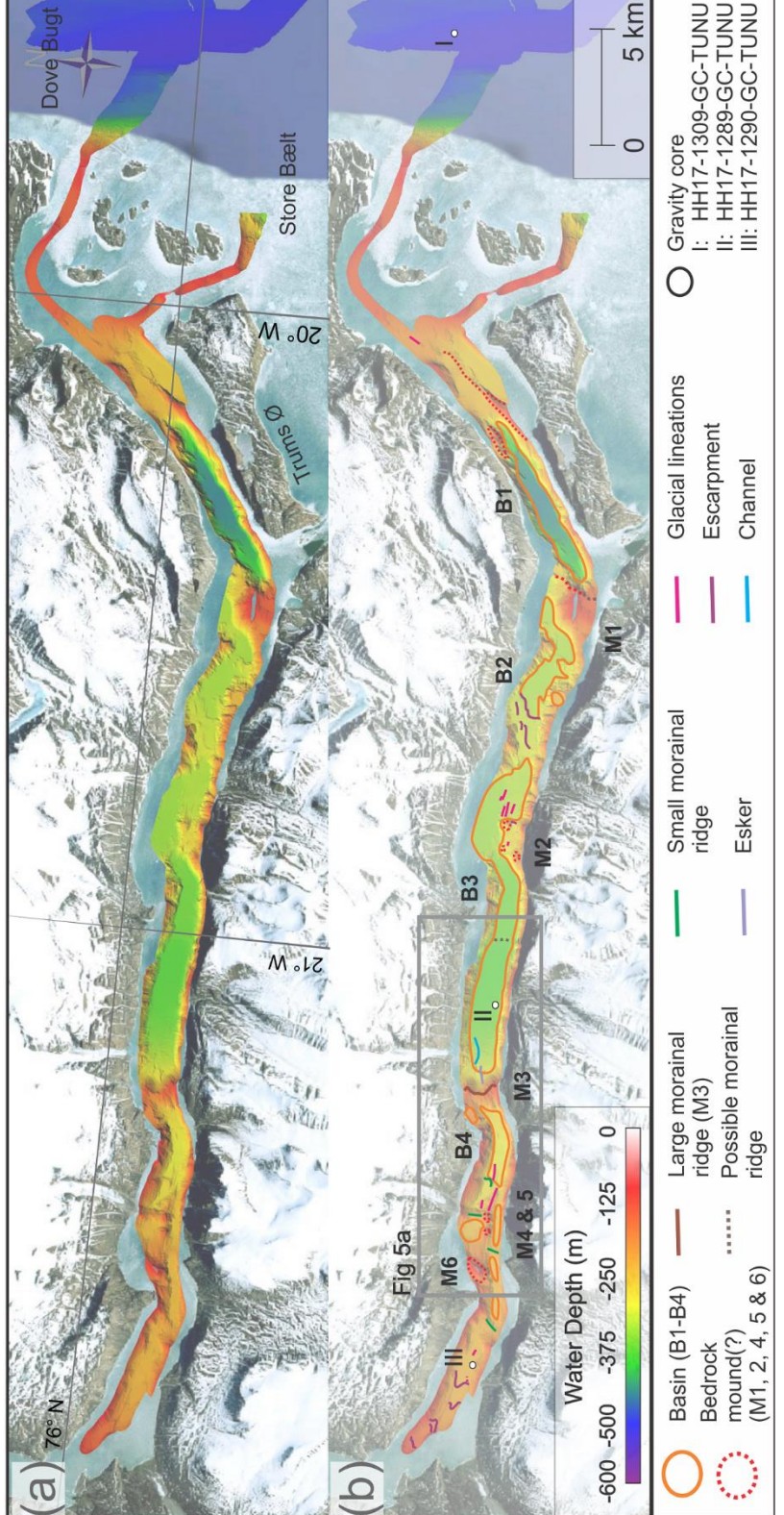

Figure 5. (a) Bathymetric map of Bessel Fjord. (b) A map of mapped features in Bessel Fjord. Satellite images obtained from Google Earth (© Google 2020).

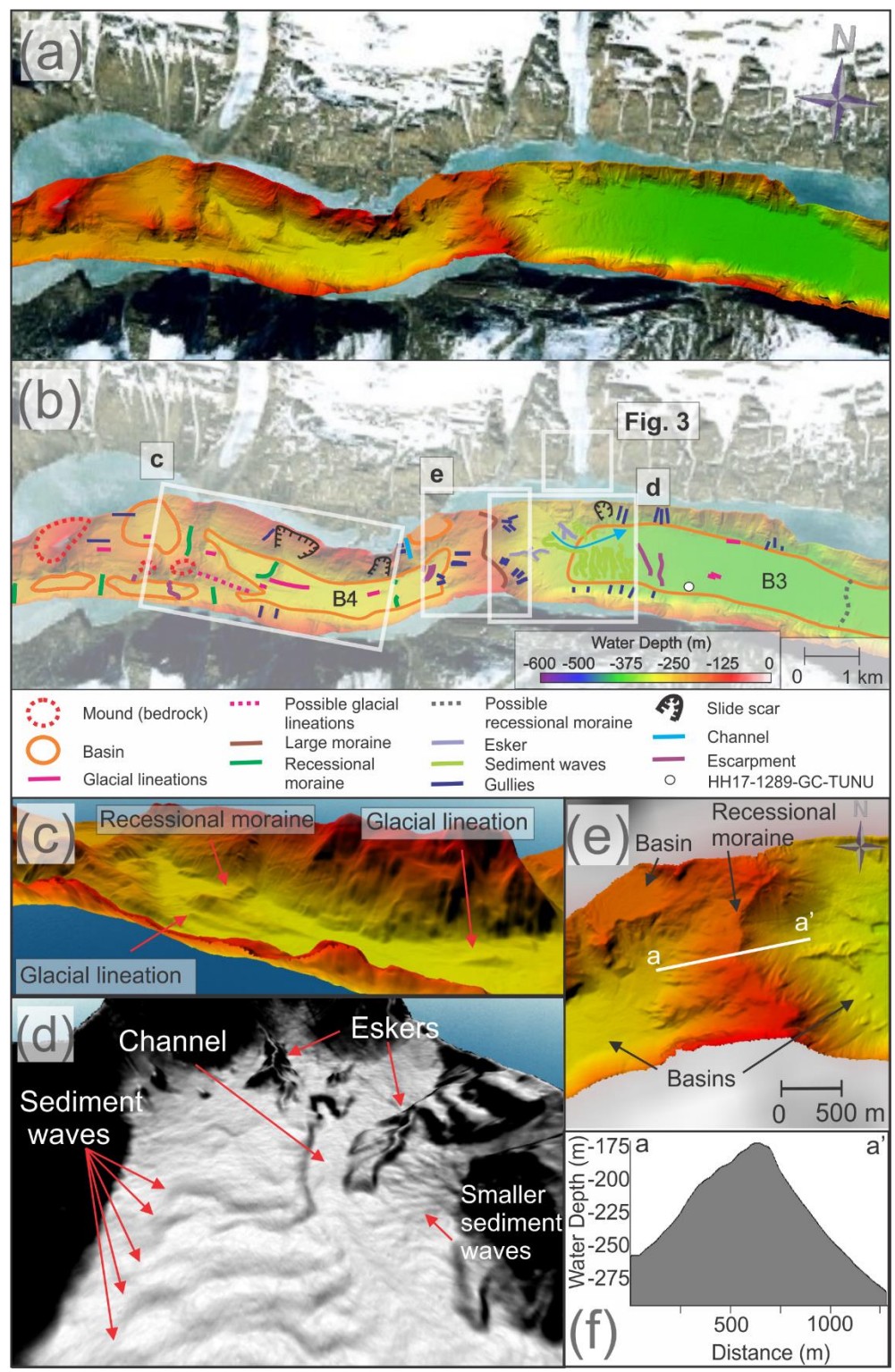

386

Figure 6. (a-b) Mapped sections from inner to middle Bessel Fjord. Background images used for 6a & 6b obtained from Google Earth (© Google 2020). (c) Glacial lineations in Basin 4 (B4). (d) Eskers, sediment waves and a channel in Basin 3 (B3). (e) A large moraine (M3) between B3 and B4. Note the raised sub-basin to the west and esker to the east. (f) Profile across the large recession moraine (M3).

(Lyså & Vorren, 1997; Dowdeswell et al., 2016) and have been observed with a variety of sizes
and morphologies on the NE Greenland shelf (e.g., Winkelmann et al., 2010).

### 4.2.4. Sinuous Ridges- Eskers

Sinuous ridges, oriented parallel or oblique to the fjord's axis, occur in basin 3 (Figs. 5, 6b, 6d
&6e). These features have widths and lengths of 50 to 120 m, 350 to 800 m, respectively and
heights of 10 to 15 m. The most pronounced examples of these ridges have been observed east
of the large recessional moraine that has been previously discussed (Fig. 6e).
These sinuous ridges have been interpreted as eskers. These landforms form from sediment
infill of subglacial and englacial conduits and have been identified in other studies in Greenland
(Huddart and Lister, 1981; Geirsdóttir et al., 2000; Winkelmann et al., 2010; Lane et al., 2015).
They frequently form in the direction of former ice flow and often form during terminal stages of
glaciation, and are therefore associated with moraines (Shreve, 1985). They vary in size
depending on the glacial drainage pattern, as well as a number of other factors, however eskers
identified within Bessel Fjord appear smaller than those identified in studies in Canada, the UK
and Kola Peninsula in Russia (Storrar et al., 2014).

### 4.2.5. Wavy Transverse Ridges- Sediment Waves

Adjacent to the two eskers in Basin 3 are a series of wavy transverse ridges to the east of a
large recessional moraine (Figs. 5, 6b & 6d). These features occupy an area of ~500 by 1500 m
and contain small ridges and flat areas that slope at an angle of 3 to 6° to the east. Each wave
"crest" is ~50 to 100 m apart, although some appear to begin only halfway through the width of
the area, where others occupy the entire width, north to south. These waves are crosscut by a
channel to the north (Fig. 6d). North of this channel similar features with a wavy morphology
occur, although these are substantially smaller.
These wavy transverse ridges have been interpreted as sediment waves. Sediment waves
found associated with deltaic and glacifluvial deltaic systems have been associated with
retrogressive slope failures, gravity-induced sediment creep and/or the migration of sediment
waves upslope (Cartigny et al., 2011; Hill, 2012; Stacey and Hill, 2016). Alternatively, given the
position of the smaller wavy transverse ridges to the ice cap on Ad. S. Jensen Land (Figs. 1 &
2) and the larger ridges to the large moraine to the west (Figs. 5 & 6) it is also possible that
these ridges are sets of moraines. Recessional moraines have been identified in the vicinity of
eskers in Spitsbergen fjords (Ottesen et al., 2008; Kempf et al., 2013), which may account for
the smaller wavy transverse ridges. The larger wavy transverse ridge do also resemble thrust
moraines identified by Forwick et al. (2010). Further work may be required in the evaluation of
these features. For a full list of observed landforms see Table 4.

### 4.3. Lithostratigraphy

Three gravity cores were retrieved from the study area. Gravity core HH17-1309 was collected
in Store Bælt and was sampled from a N/NW-S/SE oriented depression that contains iceberg
ploughmarks and a MSGL. Gravity core HH17-1289 was collected in the middle of the Bessel
Fjord and is located directly east of the above-mentioned sediment waves on the distal part of
the pronounced transverse ridge. Nearby, a modern ice cap fed glacifluvial channel is observed
in satellite imagery, likely with a delta at its fjord termination. The gravity core HH17-1290 was
collected within the inner fjord, west of the basins and thresholds observed in this study area
and is the closest core to Soranerbræen (located ~9.7 km east of the glacier) (Fig. 7).

Table 4. Overview of observed landforms in southern Dove Bugt and Bessel Fjord.

| Region | Description | Width | Length | Height | Notable Feature | Interpretation |
|---|---|---|---|---|---|---|
| **Dove Bugt** | Elongated lineations | 35-50 m | ~1->10 km | <1-3 m | Roughly N-S | Glacial Lineations |
| | *Wide | 200-650 m | 3.8 to 8.8. km | 4.5-15 m | | |
| | Depression and mound | 200 m | 450 m | 3-4 m | Mound to the south of the depression | Hill-hole pair |
| | Furrows (scour marks) | ~40-100 m | <100-200 | 3-5 m | Irregular | Iceberg plough marks |
| | Transverse ridges | 150-400 m | ~30-100 m | 0.5-1 m | Roughly W-E | Recessional moraines |
| **Bessel Fjord** | Linear ridge | 45-350 m | 100-1000 m | 3-9, 80 m | Parallel to the fjords axis | Glacial Lineations |
| | Transverse ridges | 150-600 m | 120- 500 m | <5-58 m | Perpendicular to the fjords axis | Recessional moraines |
| | *Large ridge (M3) | 1485 m | 600-1600 m | 72 to 162 m | | Moraine |
| | Sinuous ridges | 50-120 m | 350-800 m | 10-15 m | | Esker |
| | Wavy transverse ridges | 400-700 m | ~45-100 m | 2-5 m | Perpendicular to the fjords axis | Sediment wave |
| | Elongated depression | ~200 m | ~1 km | 6-8 m | | Channels |
| | Chute | ~20-100 m | 60-400 m | 1-15 m | | Gullies |

### 4.3.1. Facies

*Facies 1 – Laminated Mud (Fl, Fl-d & Fl/m-d)*

Facies 1 consists of laminated mud (Fl) and laminated mud with dropstones (Fl-d) and have been observed in all three gravity cores (Figs. 7, 8a, 8d & 8f)). Laminations are composed of either mud or very fine sand. Mud laminations with finer laminations have also been identified in Unit 3.2 (100-200 cm; Fig. 7a, Fl/m-d). Microfractures have also been identified within this facies (Fig. 8f).

Wet-bulk density measurements tend to increase with depth in some sections of this facies (e.g., 87-350 cm in HH17-1309), suggesting normal sediment consolidation. However, a stagnation or decrease in wet-bulk density with depth in other sections (e.g., below ~350 cm in HH17-1309) suggests less consolidation. The magnetic susceptibility generally tends to increase with depth in HH17-1309 and in Unit 3.2 in HH17-1290, however the remainder of this facies in HH17-1290 (Unit 3.1) remains relatively stable to the base of the core. Notable positive peaks have been identified at 110 and 140 cm in HH17-1309 and measurement fluctuations occur in HH17-1289. Peaks in magnetic susceptibility may reflect the introduction of turbidites or clasts where fluctuations may reflect shifts in sediment provenance.

Muds with sand laminations are believed to have formed through a combination of ice-proximal suspension settling from overflow plumes and turbidity-current activity (underflows). The rhythmically laminated muds are believed to have formed from ice-proximal suspension settling from turbid overflow plumes. Similar laminated sediments have been identified in Kejser Franz Joseph Fjord and Fosters Bugt in East Greenland and are theorized to have been deposited

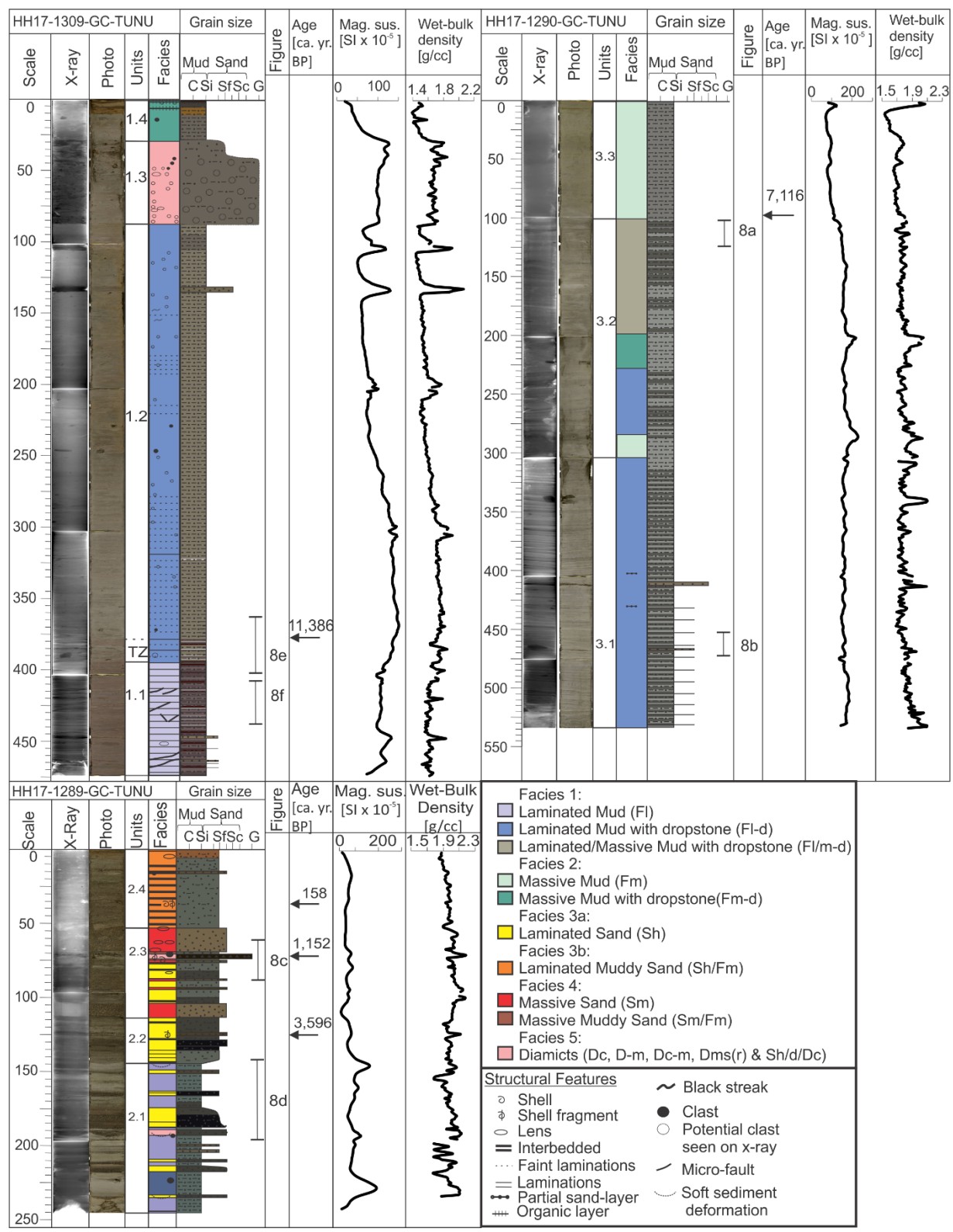


*Figure 7. Lithological core logs of the three gravity cores with x-ray images, core photos, unit divisions, facies,*
*structures, magnetic susceptibility and wet-bulk density. TZ in HH17-1309-GC-TUNU stands for "Transition Zone".*
*Grain size abbreviations: C: clay, Si: silt, Sf: fine grained sand, Sc: coarse grained sand and G: gravel.*

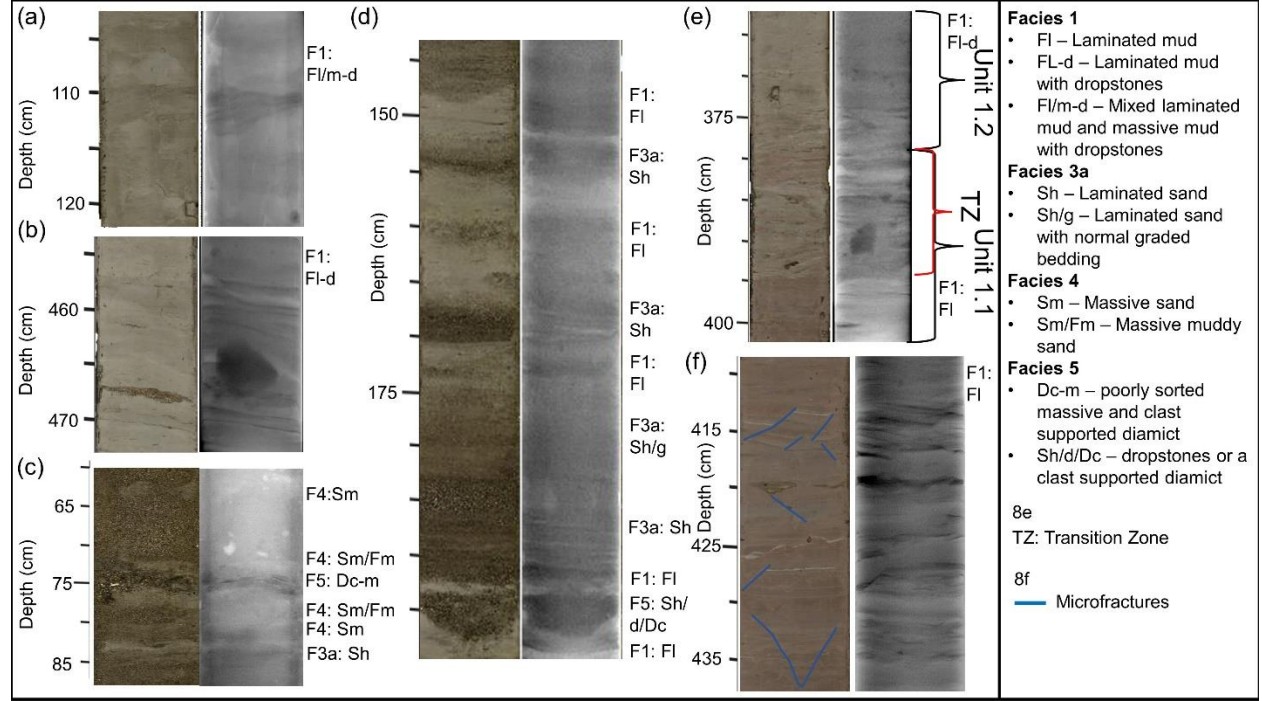

*Figure 8. Photographic and x-ray images of sections of the three gravity cores (a-f). Corresponding facies codes can*
*be found to the right of each image.*
from turbid meltwater plumes in an ice-proximal environment (Evans et al., 2002). Large clasts
have been interpreted as ice rafted debris (IRD). The formation of microfractures may have
been caused by soft sediment deformation, possibly from grounded icebergs.
*Facies 2 – Massive Mud (Fm & Fm-d)*
The second facies consists of massive mud with or without dropstones and can be found in the
inner fjord core HH17-1290 and the Store Baelt core HH17-1309 (Fig. 7). In HH17-1290 this
appears downcore between sections of Facies 1 as well as in the topmost unit, Unit 3.3. The
magnetic susceptibility gradually increases downcore in this facies in Unit 3.3. Further down
core, in Unit 3.2, this facies is associated with a downwards trend in magnetic susceptivity
following peaks in measured readings. Wet bulk density values roughly mirror these trends. In
HH17-1309 massive mud units have been observed in Unit 1.4, where magnetic susceptibility
and wet bulk density values increase downcore.
This facies is interpreted as being the result of suspension settling from overflow plumes and is
believed to have been deposited in an ice-distal glacimarine environment with varying input from
IRD (i.e., Boulton & Deynoux, 1981). Sediment may be sourced from a single location (i.e.,
Soranerbræen) or more than one location (e.g., local ice caps) in an ice-distal glacimarine
environment with limited iceberg or sea-ice rafting. Massive mud deposits have also been
identified in other Greenland fjords (e.g., Ó Cofaigh et al., 2001) and it has been suggested that
they may indicate meltwater from ice- or fjord margin-distal conditions (Evans et al., 2002).
*Facies 3a – Laminated Sand (Sh)*
Facies 3a consists of sections of sand with horizontal sand laminations. This facies has been
predominantly observed in the mid-fjord core, HH17-1289-GC-TUNU (Figs. 7 & 8d). These
sections consist of fine to medium grained sand that range in thickness and colors. Occasionally
this facies also contains normal graded bedding (e.g., Fig. 8d, ~174-183 cm). This facies does
not contain uniform magnetic susceptibly or wet-bulk density readings as it has been found in
association with low and high peaks of both parameters as well as values that are near the
average for the core.
This facies is interpreted as being deposited from turbidity currents, possibly underflows that are
either sourced from glacial or non-glacial streams and slope failures. Uniform layers may
indicate a single, rapid event, where shifts in grain size and color may be the result of short-lived
fluctuations in sediment input. Laminated sands have been identified in Scoresby Sund in East
Greenland and have also been attributed to turbidite formation (Ó Cofaigh et al., 2001).
*Facies 3b – Laminated Muddy Sand (Sh/Fm)*
Facies 3b represents sections of sand with faint horizontal laminations as well as a large
quantity of clay material interspersed throughout with faint laminations. This has been observed
in HH17-1289 at the topmost unit in the core, Unit 2.4 (Fig. 7). Magnetic susceptibility is
relatively uniform in this facies, where the wet-bulk density tends to decrease up core. Sediment
grain size analysis of a single sample from this facies revealed that the sediment is composed
of 56.3% sand and 43.7% mud. A "patch" of black organic material (i.e., plant material and
shells) was also identified within this unit.
This complex facies is believed to have formed predominantly from underflow events, sandy –
muddy turbidites, alternatively sandy turbidites with additional input from suspension settling.
Similar deposits have been observed in Balsfjord, Norway although without lamination and
possibly a higher mud content (Forwick and Vorren, 1998).
*Facies 4 – Massive Sand / Massive Muddy Sand (Sm & Sm/Fm)*
Facies 4 contains sections of massive sand (Sm) as well as massive sand with a large amount
of clay content (Sm/Fm). This facies is predominantly found in Unit 2.3 (and to a much less
extent, Unit 2.4) in HH17-1289 (Fig. 7). Sections of massive sand have been found in
association with mud lenses and often contain horizontal sand layers (Sh) above and below it.
Slight increases and decreases in magnetic susceptibility values have been observed within this
facies.
This facies is believed to have developed through rapid deposition as well as deformation of
Facies 3a & b. According to this interpretation, the mud lenses observed in this facies were
once layers/lamina that became deformed due to the sand – mud density contrast. Massive
sand has been found in Kangerlussuaq and Miki Fjords in East Greenland (Smith and Andrews,
2000) and well-sorted coarse grain deposits have been recovered near Petermann Glacier in
northern Greenland (Reilly et al., 2019). Authors have attributed these layers to sediment gravity
flows.
*Facies 5 – Diamicts (Dc, D-m, Dc-m, Dms(r) & Sh/d/Dc)*
Facies 6 contains a variety of different diamicts observed within the mid-fjord core HH17-1289
and the Store Baelt core HH17-1309. In HH17-1289 this includes a 3.5 cm poorly sorted
massive and clast supported diamict (Dc-m) in the middle of Unit 2.3 (Figs. 7 & 8c), and a
horizontally laminated layer of sand that that is either accompanied by dropstones or a clast

supported diamict (Sh/d/Dc) (Figs. 7 & 8d). It is inferred that they are the result of sea ice or iceberg rafting/dumping. Within HH17-1309 there is a substantially larger, sharp based, matrix-supported diamict, stratified in its upper part (Dms(r)) in Unit 1.3 (Fig. 7). Based on these characteristics, this diamict has been interpreted as a density flow deposit, likely a debris flow deposit that is overlain by (part of) a turbidite.

### *4.3.2. Chronology and sedimentation rates*

Shell and shell fragments were recovered from HH17-1289 for radiocarbon dating. At 34 cm depth, a semi-spherical path of organic content was identified, containing two intact *Yoldiella lenticula*, a shell fragment and plant material. Additionally, at 71 cm depth, a large 3 cm half of a *Hiatella arctica* shell was collected for dating, and shell fragments were recovered from a depth of 125 cm for the same purpose. These shells yielded radiocarbon ages of 158, 1,152 and 3,596 cal yr. BP, respectively (Table 5).

Cores HH17-1290 and HH17-1309 were subsampled for foraminifera material at four positions. Calcareous benthic species, which were rare, were used for dating and include predominantly *Melonis barleeanus* as well as *Islandiella norcrossi*, but in substantially smaller quantities. In HH17-1309, at a depth of 377 cm *Islandiella norcrossi* (rare to common) & *Stainforthia feylingi* (rare) and a planktonic species were identified immediately above the transition zone between deformed (below) and undeformed sediments (above). Radiocarbon dates for the HH17-1309 sample yielded an age of 11,386 cal yr. BP where the sample from HH17-1290 yielded an age of 7,116 cal yr. BP (Table 5).

*Table 5. Radiocarbon dates, calibrated dates, and associated linear sedimentation rates.*

| Coring station | Sampling Depth [cm] | Lab nr. | Species | $^{14}$C age BP | Marine20 cal BP (1σ range) | Marine20 cal BP | Linear sedimentation interval [cm] | Linear sedimentation rate Marine20 [cm/ka] |
|---|---|---|---|---|---|---|---|---|
| HH17-1309-GC-TUNU | 377 | 5157.1.1 | Mixed benthic foraminifera | 10357 ± 95 | 11201 - 11553 | 11386 | 0-377 | 33.11 |
| HH17-1289-GC-TUNU | 35 | 5154.1.1 | *Yoldiella lenticula* | 688 ± 34 | 61 - 253 | 158 | 0-35 | 221.52 |
| HH17-1289-GC-TUNU | 71 | 5155.1.1 | *Hiatella arctica* | 1747 ± 28 | 1065 - 1250 | 1152 | 35-71 | 31.25 |
| HH17-1289-GC-TUNU | 125.5 | 5156.1.1 | Bivalve frag. | 3809 ± 36 | 3472 - 3701 | 3596 | 71-125.5 | 15.16 |
| HH17-1290-GC-TUNU | 97 | 5158.1.1 | Mixed benthic foraminifera | 6800 ± 80 | 6990-7250 | 7116 | 0-97 | 13.63 |

Linear sedimentation rates were calculated assuming modern sediments are at the core top as no overpenetration was recorded during the sampling of these cores and that during the core logging little sediment disturbance was found (Table 5). Given the scarcity of biological material in these cores these sedimentation rates act only as a first approximation until a more detailed record can be recovered. Using the available (calibrated) dating results, sedimentation rates of ~15 cm/ka, ~31 cm/ka, & ~222 cm/ka were calculated for core HH17-1289 at 71-125.5 cm, 35-71 cm, and 0-35 cm, respectively. These results reveal an increase in the sedimentation rate towards the present. However, as this core includes multiple deposits from turbidity currents

(i.e., reworked deposits), linear sedimentation rates in core HH17-1289 should be treated with
caution. An average, linear rate of ~14 cm/ka was calculated for the interval of 0-97 cm in core
HH17-1290 and an average, linear rate of ~33 cm/ka was also obtained for the large interval of
0-377 cm in core HH17-1309. These linear rates are lower, up to an order of magnitude, when
compared to the Kejser Franz Josef Fjord system ~400 km south of the study area (Olsen et al.,
2022). The origin of this observed difference must await further studies.

## 565 5. Discussion

### 566 5.1. Ice Sheet advance

The appearance of glacial lineations in Bessel Fjord suggest that the fjord was once fully
glaciated, which is in accordance with the inferred shelf break-terminating ice sheet inferred for
the LGM from other studies (e.g., Laberg et al., 2017; Olsen et al., 2020) (Figs. 9a & 9b). Ice
that filled the fjord is believed to most likely be from the modern Soranerbræen glacier but may
have also included ice caps and other nearby branches of inland ice.
Glacial lineations are believed to have formed during the LGM but could have also formed
during an ice readvance in the deglaciation (see below). Onshore and south of Bessel Fjord,
two sets of striations identified in Langsødalen (Hjort, 1979, 1981) may suggest that this valley
experienced two glaciation events (Fig. 1c). Striations, and lateral moraines, found along the
fjord axis may be the result of the east-west movement of ice through the valley, where SW
oriented striations may be the result of Storstrømmen encroaching also onto terrestrial areas.
Hjort (1981) suggested that striae on Haystack may indicate that ice flow was dominant from the
north during the Nanok Stadial but ice pressure from Langsødalen dominated later after
deglaciation begun. Thus, it is possible that ice masses drained through both Bessel Fjord and
Langsødalen during full-glacial conditions further advancing into Dove Bugt/Store Bælt.
In Store Bælt, the orientation of glacial lineations (e.g., MSGLs) suggest that ice flowed to the
south along the west cost of Store Koldewey, marking the southwards expansion of the
Storstrømmen ice stream (Figs. 9a & 9b). East of Dove Bugt, MSGLs identified in Store
Koldewey Trough are believed to have formed when the Storstrømmen ice stream acted as a
"pure" ice stream (Bentley, 1987; Stokes and Clark, 1999) and overrode the underlying
topography during the LGM (Fig. 9a; Olsen et al., 2020). It was theorized that at a later phase,
when the ice sheet began to thin, the ice stream became more influenced by the topography of
deep troughs, draining northwards to Jøkelbugten and southwards to Dove Bugt (Olsen et al.,
2020). Assuming these two phases occurred in the Storstrømmen ice stream development, it is
possible that these glacial lineations in Store Bælt represent a period when a branch of the ice
stream began conforming to topographical controls (e.g., Store Koldewey) and flowed towards
the south. At this point the ice may have flowed into the southeast through Dove Bugt Trough
(Fig. 9a).
An alternative interpretation, that cannot be excluded, is that these MSGLs formed during a
glacial re-advance that followed the LGM. Between Hochstetter Forland and Shannon Ø a
submerged moraine has been identified in Shannon Sound, which may indicate that at one point
the ice stream travelled south rather than through Dove Bugt Trough (Figs. 9b & 10a; Hjort,
1981; Landvik, 1994; Larsen et al., 2016; Funder et al., 2021). However, onshore deglaciation
ages in Store Koldewey, Germania Land and Trums Ø, do not support an ice advance during
the Younger Dryas (Fig. 10b; see below). This was possibly an ice readvance of the GrIS
outlet(s) (Soranerbræen, L. Bistrup Bræ and/or Storstrømmen) through western, inner Dove

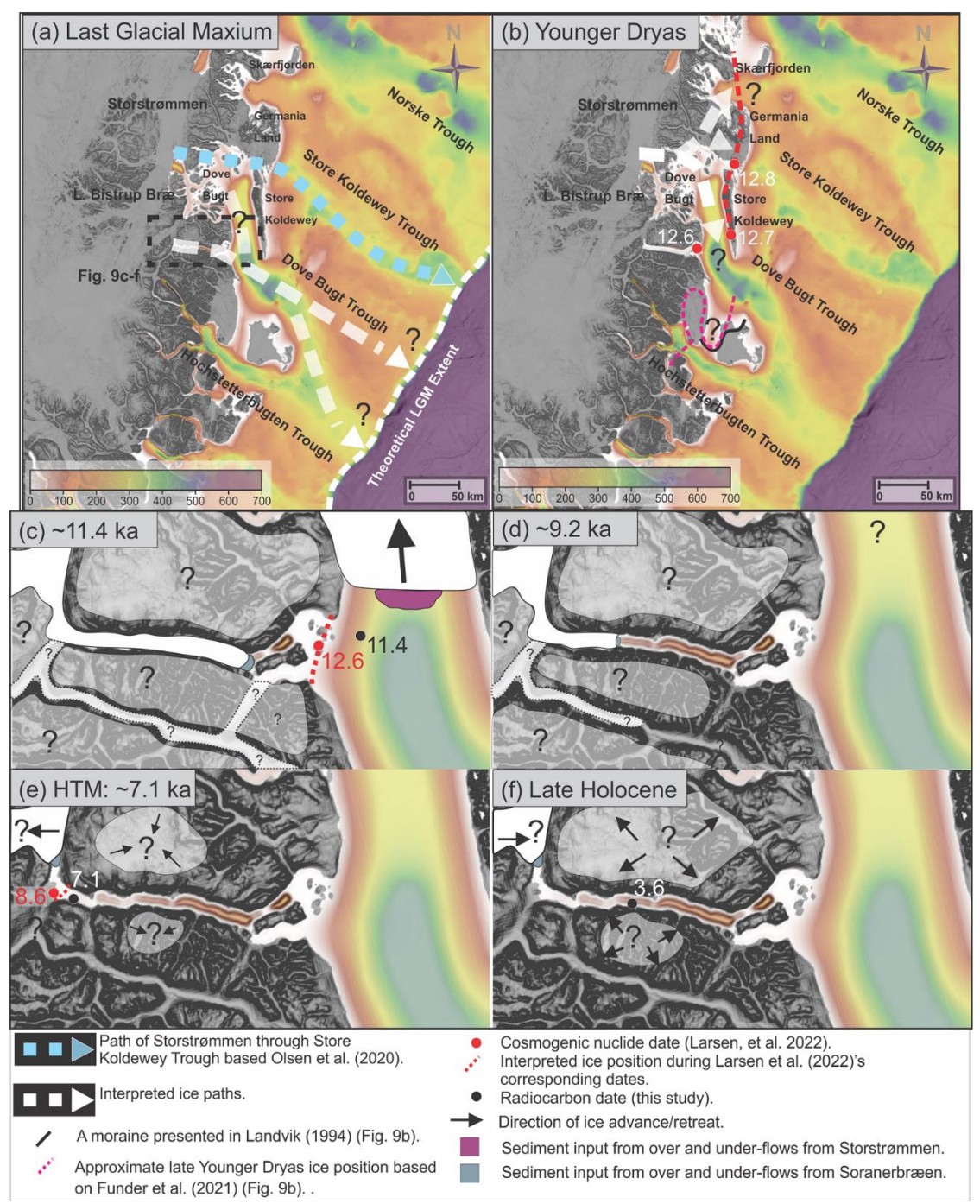


*Figure 9. Maps showing ice sheet extent and advancement/retreat directions in SW Dove Bugt and Bessel Fjord*
*during a range of periods. (a) The interpreted position of the ice sheet during the LGM. (b) The theoretical position of*
*ice in Bessel Fjord and Dove Bugt during the Younger Dryas. (c) The ice position in Bessel Fjord at ~11.4 ka based*
*on approximated deglaciation date presented in this study and the position and radiocarbon date for gravity core*
*HH17-1309. The size of ice caps in c-f are only indicative. (d) The position of ice in Bessel Fjord at ~9.2 based on*
*approximated deglaciation data from this study. (e) Ice retreating beyond our gravity core (HH17-1290) at ~7.1 ka*
*during the HTM. (f) The Late Holocene ice expanse in Bessel Fjord with a radiocarbon date from gravity core HH17-*
*1289. Background bathymetry displayed using IBCAO data (Jakobsson et al., 2020).*

612

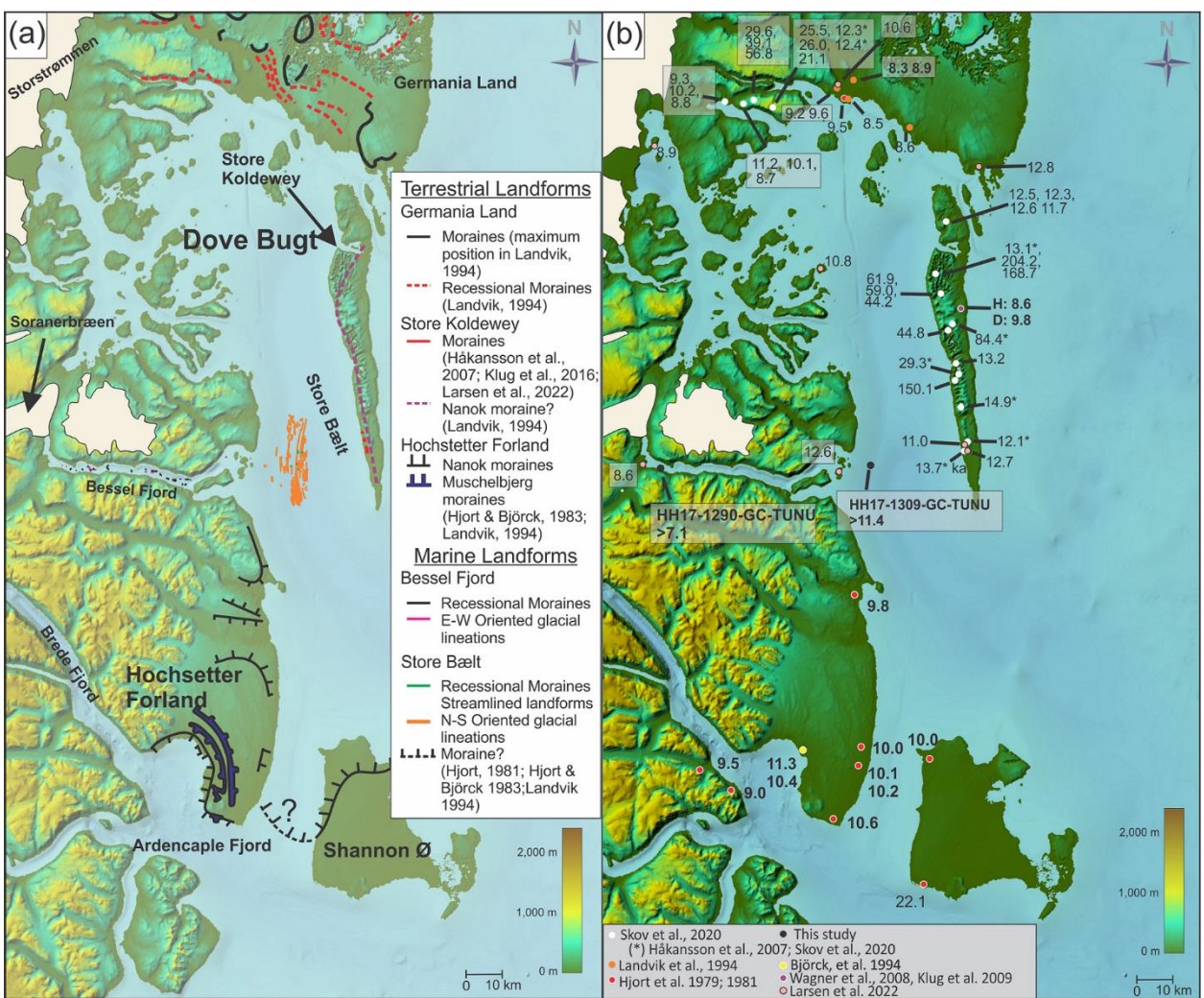

Figure 10. (a) Marine moraine ridges and glacial lineations from the current study together with previously mapped
marine and terrestrial features. (b) Location of deglaciation dates from this study (Table 5) and previous publications.
See Table 3 for recalibrated radiocarbon dates. H: Hjort Lake, D: Duck Lake. Background displayed using IBCAO
data (Jakobsson et al., 2020).

Bugt (Fig. 9b), where the surroundings (onshore and offshore) were not or less affected. If this
is correct, the readvance may have occurred during the Younger Dryas (prior to 11.4 ka cal BP,
see below).

### 5.2.    Ice Sheet retreat through Store Bælt

The deglaciation age of 11.4 ka cal BP (Table 5) from Store Bælt immediately east of the Bessel
fjord entrance is attributed to the retreat of a N-S bound branch of the NEGIS (Fig. 9c) due to
the presence of N-S oriented glacial lineations near the gravity core. This date represents a
minimum age for the deglaciation as it is not from the base of the deglacial deposits. Previously
published dates constraining the timing of deglaciation in Dove Bugt have been restricted to
terrestrial regions (Fig. 10b). Using cosmogenic nuclide dating, Skov et al., (2020) produced
deglaciation ages of ca. 12.7 ka cal BP at Store Koldewey and ca. 9.8 ka cal BP at Pusterdal
and later Larsen et al. (2022) produced a number of deglaciation ages across Dove Bugt and
Bessel Fjord (8.6-12.8 ka cal BP) (Fig. 10b).

Our minimum age of ~11.4 ka cal BP from HH17-1309 largely matches findings in Dove Bugt
and Hochstetter Forland (Fig. 10b). It is slightly later than cosmogenic nuclide ages obtained
from Larsen et al. (2022) on Trums Ø (12.6 ka cal BP) and a Nanok moraine on southern Store
Koldewey (12.7 ka cal BP), but earlier than a second Store Koldewey Nanok moraine (11.0 ka
cal BP) as well as positions closer to the modern ice margin of Storstrømmen, such as Licht Ø
(10.8 ka cal BP) and Bræ Øerne (8.9 ka cal BP). Thus, Store Koldewey, and Trums Ø may have
been partially deglaciated slightly prior to the final retreat of the NEGIS through Store Bælt.
Radiocarbon dates obtained from lake sediments on Store Koldewey suggest that the earliest
onset of warmth may have begun ~10 ka cal BP (Klug et al., 2009), therefore, the deglaciation
of the area beginning prior to this may further support these results. Additionally, Landvik (1994)
produced a range of deglaciation ages between 9.6 to 8.5 ka cal BP along the northern coast of
Dove Bugt (Hvalrosodden and Snenæs on Germania Land) and Hjort (1981, 1979) provided a
range of delegation ages between 10.6 to 9.8 ka cal BP on Hochstetter Forland. Later Björck et
al. (1994), on Hochstetter Forland,  dated *Hiatella arctica* shells near the shore of Peters Bugt
Sø and *Portlandia arctica* shells in a delta distal to a Nanok I ridge to 10.4 and 11.3 ka cal BP,
respectively (Table 3; Fig. 10b).
Although based on a limited data set, the lack of prominent morainal landforms in Store Bælt
may also suggest a rapid retreat through the region. A small number of retreat moraines have
been observed in an isolated region of the study area, but the most prominent geomorphic
landforms are glacial lineations. Placing Store Bælt within the context of Dowdeswell et al.
(2008)'s proposed model for ice streams in high latitudes, ice likely retreated through the area
rapidly, although the presence of small moraines may suggest brief periods of stagnation. This
is in accordance with findings by Larsen et al. (2020, 2022) that deep fjords and outer regions in
eastern North Greenland were rapidly deglaciated between ~12.6 and 10 ka cal BP. However,
additional data is required to confirm this.
Oceanic warming is believed to have contributed to the deglaciation of the inner shelf further
north and south of Dove Bugt (e.g., Jackson et al., 2022; Davies et al., 2022). Within the study
area, Store Koldewey does largely block oceanic water from the shelf from entering Store Bælt,
however, it is possible that warmer water traveled through the Dove Bugt Trough to the south
and impacted a north-south branch of the ice stream. This mechanism for warm water transport
has also been suggested for other east Greenland troughs (Arndt et al., 2015) and used to
explain how warm water has reached other outlets of the NEGIS (e.g., Zachariae Isstrøm via
the Norske Trough (Schaffer et al., 2017)).
*5.3.      Ice Sheet retreat through Bessel Fjord*
Cosmogenic nuclide dates from Trums Ø suggest that the deglaciation of the outer fjord began
around 12.6 ka cal BP. Gravity core HH17-1290, collected from the inner fjord region, consists
of sediments that reflect an increasingly ice distal environment up core. One radiocarbon date
from the core provides a minimum age of ~7.1 ka cal BP for the deglaciation of Soranerbræen
and/or local ice caps from the inner fjord region (Table 5 & Fig. 9e). This date, however, is not
from the base of the deglacial deposits and therefore represents a minimum age for the
deglaciation of the inner fjord. New cosmogenic nuclide dates from Vandrepasset (onshore
innermost Bessel fjord area, connecting the fjord and the next valley to the south) provide an
age of 8.6 ka cal BP for the deglaciation of the innermost fjord area (Larsen et al., 2022),
confirming this interpretation. Our minimum age of 7.1 ka cal BP and the results of Larsen
(2022) falls within a modelled ice sheet extent by Lecavalier et al. (2014) which placed the
position of the ice sheet in the middle of Bessel Fjord at 9 ka cal BP and that the present-day ice
margin is reached by 6 ka cal BP. The minimum age also agrees with the onset of HTM on
Store Koldewey (~8.0 to 4.0 ka cal BP) (Wagner et al., 2008; Klug et al., 2009; Schmidt et al.,
2011) and Hochstetter Forland (8.8 and 5.6 ka cal BP) (Björck & Persson, 1981; Björck et al.,
1994). Thus, the GrIS retreated from the marine realm in early Holocene, slightly before or at
the time of the HTM in this region (characterized by a mean July temperature 2-3°C higher than
at present; Bennike et al., 2008).
The appearance of recessional moraines in Bessel Fjord suggests that the fjord underwent a
stepwise deglaciation. The large moraine identified between Basin 3 and Basin 4 (M3; Fig. 7e)
is believed to have formed during a major ice halt or readvance, possibly climatically induced.
Smaller moraines occasionally follow topographic boundaries, which may suggest that the
retreat of ice in Bessel Fjord was also partly topographically controlled. Recessional moraines
identied by Olsen et al (2020) east of Dove Bugt in Store Koldewey Trough contain similar
heights to those identified here (excluding M3). However, there are more moraines identified in
Store Koldey Trough than in Bessel Fjord, and they are wider, which is likely due to the lack of
topographic confinment.
A decrease in atmospheric temperatures in early Holocene is recorded in the Greenland
Summit temperature records and includes the Preboreal Oscillation, the 9.2 ka event, the Pre-
8.2 ka cooling, and the 8.2 ka event, with the 8.2 ka event being the largest hemispheric-wide
negative temperature excursion during the Holocene (Kobashi et al., 2017). We tentatively
suggest that some of the moraines identified in the Bessel fjord may have developed during
some of these events. From this we suggest that increased Northern Hemisphere summer
insolation that peaked in the early Holocene was the main control for this part of the deglaciation
during which the ice front receded from the coastline to the west of (onshore) Bessel Fjord, a
distance of ~60 km. Assuming that this occurred over a maximum period of ~4.3 ka cal BP
(11.4-7.1 ka cal BP, see discussion above on the timing and length of this period), this
corresponds to an average ice recession rate of ~14 m/yr. This rate, a minimum rate, is
considered realistic as it is half (or less) than the rate estimated from the Nioghalvfjerdsfjorden
further north (also part of the Storstrømmen ice stream) where a rate of ~30-40 m/yr was
reported (Bennike & Björck, 2002).
Applying this minimum rate to the distance between Trums Ø (Larsen, et al., (2022); 12.6 ka cal
BP) and the major mounds and moraines identified in this study (M1, M2, M3 & M6), yields the
approximate minimum ages of 11.4, 10.5, 9.7 and 9.2 ka, respectively. This places
Soranerbræen between large moraine M3 and the bedrock mound M6 around the 9.2 ka event
(Fig. 9d). This is noteworthy as M3, and other many of the smaller moraines identified between
these two features, may have formed during this climatically cooler period. Additionally, many
smaller moraines in the fjord follow topographic boundaries, which may suggest that the retreat
of ice in Bessel Fjord was partly topographically controlled.
While oceanic warming may be partially responsible for the retreat of the NEGIS through Store
Bælt, we believe that Bessel Fjord is too sheltered by the sill at its entrance to have allowed
warm, intermediate water to enter and make a significant impact of the deglaciation of the
southern outlet of Soranerbræen. Our bathymetric dataset reveals that the depth of the sill is
between ~50 to 200 m, however large parts of it are above water and form islands. This is far
shallower than other fjord sills in the region that are theorized to have blocked warm Atlantic
Water (e.g., the sill in Dijmphna Sund to the north, which has a maximum depth of 170 m
(Wilson and Straneo, 2015)). Also, the effect of the glacio-eustatic readjustment is considered to
be small for this region, ~9.5 m higher in the Young Sound region (slightly south of our study
area) 7500 years ago (Pedersen et al., 2011). Rignot et al. (2022) also theorized that seafloor
topography may impact whether warm water is reaching the northern outlet of Soranerbræen.
They suggested further that the grounding line retreat of Storstrømmen, L. Bistrup Bræ,and
possibly Soranerbræen, may primarily be caused by ice thinning from atmospheric warming
(Rignot et al., 2022). We suggest that a similar mechanism may be responsible for
Soranerbræen's retreat through Bessel fjord during the deglaciation.
*5.4.       Holocene glacier variability and sedimentary processes in Dove Bugt*
Sedimentological evidence (e.g., laminated muds) from HH17-1309 suggests, that suspension
settling from a glacial source(s) likely dominated southwestern Dove Bugt during the Holocene.
The contribution of sediment from the NEGIS seems unlikely, as Pusterdal became deglaciated
by 9.5 ka cal BP (Skov et al., 2020) and Storstrømmen retreated beyond Bræ Øerne by 8.9 ka
cal BP (Larsen et al., 2022), therefore it very well may be from Soranerbræen, or local ice caps.
During the latter part of the HTM in the middle Holocene, a time period in which some glaciers
are believed to have reached their minimum extent across Greenland, the NEGIS is believed to
have retreated beyond its current position between 5.4 to 1.2 ka cal BP (Table 3), creating the
Storstrømmen Sound (Weidick et al., 1994). Relating the core sedimentology to a linear age
model developed from sedimentation rates (i.e., Table 5), laminations appear less frequently in
core HH17-1309 during this period, yet they are not absent. Laminations are entirely absent in
the Bessel Fjord core HH17-1290 during this period and remain absent through the colder Late
Holocene. Later, during the Little Ice Age, Storstrømmen demonstrated to have expanded to its
modern day position (Weidick et al., 1994).
Gravity core HH17-1289, collected to the north of an onshore glaciofluvial channel connected to
a modern-day ice cap, transitions to complex assortment of sand layers just prior to 3,596 cal yr
BP (Fig. 7). Sedimentological evidence suggests that these sand layers are largely the result of
rapid, short lived depositional events (i.e., turbidity currents) interpreted to be related to the
growth of a delta slightly south of the core site, from glaciofluvial sediment input from a nearby
outlet glacier.
Pollen assemblage data from Hochstetter Forland mark the end of the HTM at 5.6 yr BP (Björck
and Persson, 1981; Björck et al., 1994) and information derived from aquatic organisms mark
the end of the HTM on Store Koldewey at 4 yr BP (Wagner et al., 2008; Klug et al., 2009b;
Schmidt et al., 2011). This coincides with the onset of turbidites in core HH17-1289. Therefore,
it is possible that this shift to sand dominated sedimentation within this core was controlled by
climatically driven processes. This onset is here suggested to result from higher sediment input
through the channel as local ice caps expanded outwards following the HTM, possibly in
response to this climate cooling (Fig. 9f). This period of cooling also corresponds to extended
concentrations of sea ice on the shelf (Kolling et al., 2017).

# 6. Conclusion
In summary:
• Glacial lineations (MSGLs) identified in SW Dove Bugt suggest fast-flowing ice,
interpreted to be from the NEGIS, developed during the LGM or an ice readvance during
the deglaciation.

- Our minimum deglaciation date for Store Bælt (>11.4 ka cap BP) is slightly later than new cosmogenic nuclide dates found onshore on Trums Ø and one of two Nanok stadials on Store Koldewey (Larsen et al., 2022) as well as various other dates across Store Koldewey (e.g., Skov et al., 2020). Thus, Store Koldewey and Trums Ø may have been partially deglaciated prior to the final retreat of the NEGIS through Store Bælt.
- Moraines in Bessel Fjord (to the west of Dove Bugt) suggests that the fjord underwent multiple halts/or readvances upon deglaciation. Thus, the bathymetry of Bessel Fjord indicates that the glacial dynamics of the fjord were more dynamic than onshore evidence suggests.
- The radiocarbon date of 7.1 ka cal BP obtained in an inner fjord core is interpreted as a minimum age at which Soranerbræen retreated to or beyond its present-day onshore position west of the fjord and is in conformity with cosmogenic nuclide dates presented by Larsen et al. (2022) in the onshore inner fjord (8.6 ka cal BP).
- Ice recession in Bessel Fjord occurred at a minimum average rate of ~14 m/yr.
- The GrIS retreated from the marine realm in the early Holocene, around the time of the onset of the HTM in this region. From this we suggest that increased Northern Hemisphere summer insolation that peaked in the early Holocene was the main control for this part of the deglaciation.
- A low sedimentation rate of 13.63 cm/ka after 7.1 ka cal BP in HH17-1289, and the presence of only massive mud, suggests that Soranerbræen did not expand back into Bessel Fjord for the remainder of the Holocene.
- The transition of mud to muddy sand at 4 ka cal BP in a mid-fjord core HH17-1289 may provide evidence for local ice cap growth. Thus, ice caps in Bessel Fjord may have fluctuated with greater sensitivity to climatic conditions than the NE sector of the GrIS during the cooling phase that followed the HTM.

*Data availability:* The bathymetry and core data from UiT The Arctic University of Norway will be available upon reasonable request at UiT's open research data repository: https://dataverse.no/dataverse/uit.

*Author contributions*: Jan Sverre Laberg and Tom Arne Rydningen designed this study and collected the new data during the 2017 TUNU VII cruise. The bathymetrical and lithological data were interpreted by Kevin Zoller in collaboration with Jan Sverre Laberg and Tom Arne Rydningen. Kevin Zoller prepared the manuscript with contributions from all co-authors.

*Competing interests:* The authors declare that they have no conflict of interest.

*Acknowledgement:* We would like to thank the participants of the 2017 TUNU cruise to Greenland for making this project possible. A special thanks to the captain and crew of the RV *Helmer Hanssen* for their involvement in the cruise and assistance in collecting the data. A thanks also goes out to the lab staff at UiT, Trine Dahl, Karina Monsen and Ingvild Hald, who assisted with processing sediment core samples for this project. We would also like to thank Gesine Mollenhauer and the lab staff at the Alfred Wegener Institut for providing us with radiocarbon dated material using their MICADAS. Funding for this work was provided by UiT The Arctic University of Norway.

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
