# Peer review of "A high-Arctic inner shelf–fjord system from the Last Glacial Maximum to the Present: Bessel Fjord and SW Dove Bugt, NE Greenland"

_EGUsphere, 2022_

## Author Response (AR1)

*AC: Thank you for accepting our resubmission. We have addressed the comments from reviewers #1 and #2 below. The reviewers' original comments are in black, our old "Author Responses" prior to editing the manuscript are written in red, and new "Author Responses", written after the completion of manuscript edits, are written in green. Please refer the green text concerning the resubmission. Additionally, individual lines mentioned in the green text refer to lines from the "track-change" document*

**Reviewer #1 Comments and Responses:**

AC: Thank you for thoroughly reading our manuscript and adding insightful comments. We found your review to be very helpful and appreciate the time that you spent going through it. We believe it will help improve our manuscript. Below we have addressed each comment one by one, with the original comments in bold and labelled "RC" and our response labelled "AC".

**RC: The paper presents new and interesting multibeam and sediment core data from Bessel Fjord and Dove Bugt in NE Greenland. It is based on a Msc thesis from 2020 by the first author and the main results have been synthesized in the manuscript. This process has clearly not been easy and there are many things that could be improved. The structure is not clear, and there are many typos and redundant sentences, and it also needs to be updated with the newest scientific literature from the area. However, with some substantial editing it has the potential to become a valuable scientific contribution. My comments below should therefore not be seen as an excuse to reject the paper – rather the opposite – it should be seen as help to improve it.**

**RC: Major concerns:**

**RC: 1) Both the title and the abstract make a great deal out of this being a study of the southern part of NEGIS. It is not. The main part of the study concerns Bessel Fjord where most of the multibeam data and 2 out of the 3 marine cores are from. Only a minor part of the multibeam data and 1 sediment core is from outside the mouth of Bessel Fjord and could potentially inform about the NEGIS. Accordingly, I suggest that the authors refocus the study to account for the glaciation history in Bessel Fjord where most of data is from their data is from.**

AC: We will refocus the manuscript on Bessel Fjord and the inner Dove Bugt. This is reflected in the revised title of the manuscript (following Reviewer #2's comment).

AC: The manuscript was refocused to put a larger emphasis on Bessel Fjord and the surrounding region. This is reflected in the revised title "A high-Arctic inner shelf–fjord system from the Last Glacial Maximum to the Present: Bessel Fjord and SW Dove Bugt, NE Greenland".

**RC: 2) Several relevant papers are missing and would be very relevant to include that in the discussion:**

Pados-Dibattista, T., Pearce, C., Detlef, H., Bendtsen, J., Seidenkrantz, M.-S., 2022. Holocene palaeoceanography of the Northeast Greenland shelf. Climate of the Past 18, 103-127.

Katrine Elnegaard, H., Lorenzen, J., Davies, J., Wacker, L., Pearce, C., Seidenkrantz, M.-S., 2022. Deglacial to Mid Holocene environmental conditions on the northeastern Greenland shelf, western Fram Strait. Quaternary Science Reviews 293, 107704.

Rasmussen, T.L., Pearce, C., Andresen, K.J., Nielsen, T., Seidenkrantz, M.S., 2022. Northeast Greenland: iceâ free shelf edge at 79.4°N around the Last Glacial Maximum 25.5–17.5 ka. Boreas, bor.12593.

Jackson, R., Andreasen, N., Oksman, M., Andersen, T.J., Pearce, C., Seidenkrantz, M.-S., Ribeiro, S., 2022. Marine conditions and development of the Sirius Water polynya on the North-East Greenland shelf during the Younger Dryas-Holocene. Quaternary Science Reviews 291, 107647.

Davies, J., Mathiasen, A.M., Kristiansen, K., Hansen, K.E., Wacker, L., Alstrup, A.K.O., Munk, O.L., Pearce, C., Seidenkrantz, M.-S., 2022. Linkages between ocean circulation and the Northeast Greenland Ice Stream in the Early Holocene. Quaternary Science Reviews 286, 107530.

Larsen, N.K., Søndergaard, A.S., Levy, L.B., Strunk, A., Skov, D.S., Bjørk, A., Khan, S.A., Olsen, J., 2022. Late glacial and Holocene glaciation history of North and Northeast Greenland. Arctic, Antarctic, and Alpine Research 54, 294-313.

Smith, J.A., Callard, L., Bentley, M.J., Jamieson, S.S.R., Sánchez-Montes, M.L., Lane, T.P., Lloyd, J.M., McClymont, E.L., Darvill, C.M., Rea, B.R., O'Cofaigh, C., Gulliver, P., Ehrmann, W., Jones, R.S., Roberts, D.H., 2022. Holocene history of 79° N ice shelf reconstructed from epishelf lake and uplifted glacimarine sediments. Ice sheets/Paleo-Glaciology (including Former Ice Reconstructions). The Cryosphere

Håkansson, L., Graf, A., Strasky, S., Ivy-Ochs, S., Kubik, P.W., Hjort, C., Schluchter, 2007. Cosmogenic Be-10-ages from the Store Koldewey island, NE Greenland. Geografiska Annaler Series a-Physical Geography 89A, 195-202.

*The original Author Response to the reviewers' comments have been edited here to better reflect changes in the manuscript*

AC: Thank you for providing additional publications, they will be included in the revised manuscript where relevant. This includes a paragraph summarizing the main findings from the NE Greenland shelf. Håkansson et al. (2007) is in the Manuscript (in Figure 10) however this was accidently not added to the references. This will be corrected.

There is however, one important point to keep in mind when comparing our results to the publications above. This manuscript (Zoller et al.) was originally using the MARINE13 calibration curve for 14C calibration (Reimer et al., 2013). This is because Heaton et al. (2020) in their paper presenting the newer MARINE20 curve states that "…uncertainty in past polar climate conditions can have a significant effect on calibration in these regions." From this, they conclude that "The current Marine20 curve is not therefore suitable for calibration in these polar regions." We inferred that the fjords and coastal areas of NE Greenland is included in their "Polar regions" and thus followed their recommendations. This is in accordance with Rasmussen et al. (2022) cited above (using MARINE13), but not Jackson et al. (2022), Davies et al. (2022), Hansen et al. (2022), and Pados-Dibattista et al. (2022) who are using MARINE20.

Later, in their 2022 paper, Heaton et al. recommend the use of MARINE20 also for Polar regions, given that the ages are younger than 11.5 ka BP. This is the case for our samples. Therefore, in the revised manuscript (Table 4), we use calibrated dates using MARINE20 forming the basis for our "Discussion" (Figure 10). We have also recalibrated previously published 14C ages relevant for our "Discussion".

Heaton, T. J., Köhler, P., Butzin, M., Bard, E., Reimer, R. W., Austin, W. E. N., Ramsey, C. B., Grootes, P. M., Hughen, K. A., Kromer, B., Reimer, P. J., and Heaton, T. J.: Marine20 — The Marine Radiocarbon Age Calibration Curve (0-55,000 CAL BP), Radio, 62, 779–820, https://doi.org/10.1017/RDC.2020.68, 2020.

Heaton, T. J., Bard, E., Ramsey, C. B., Butzin, M., Hatté, C., Hughen, K. A., Köhler, P., and Reimer, P. J.: A Response to Community Questions on the Marine20 Radiocarbon Age Calibration Curve: Marine Reservoir Ages and The Calibration of 14C Samples from the Oceans, Radiocarbon, 65, 247–273, https://doi.org/10.1017/RDC.2022.66, 2022.

Reimer, P. J., Bard, E., Bayliss, A., Beck, J. W., Blackwell, P. G., Bronk Ramsey, C., Buck, C. E., Cheng, H., Edwards, R. L., Friedrich, M., Grootes, P. M., Guilderson, T. P., Haflidason, H., Hajdas, I., Hatté, C., Heaton, T. J., Hoffmann, D. L., Hogg, A. G., Hughen, K. A., Kaiser, K. F., Kromer, B., Manning, S. W., Niu, M., Reimer, R. W., Richards, D. A., Scott, E. M., Southon, J. R., Staff, R. A., Turney, C. S. M., and van der Plicht, J.: Intcal13 and Marine13 Radiocarbon Age Calibration Curves 0–50,000 Years Cal Bp, Radiocarbon, 55, 1869–1887, 2013.

AC: Recommended publications (as well as other updated information) has been added to the manuscript:

The location of specific recommended publications can be found below:

Pados-Dibattista et al., 2022: line 198

Rasmussen et al., 2022: line 76

Jackson et al., 2022: line 70, 183-189, 325, 369, 818

Davies et al., 2022: line 70, 87, 198, 818

Larsen et al., 2022: line 85, 88, 209, 229, 248, 250, 256, 765, 772, 814, 845/846, 879, 907/908, 964, Fig. 9 & 10

Håkansson et al., 2007: line 249, Fig. 10

Elnegaard et al., 2022 and Smith et al., 2022 were not added to the manuscript.

**RC: 3) Description and interpretation of sediment cores**

**The description and interpretation of the sediment cores could be simplified. Overall, it seems that 2 units would be sufficient to describe the changes in palenvironmental conditions recorded in the cores i.e., proximal, and distal marine mud. Maybe a third unit comprising the sandy interval in 1289 reflecting Neoglacial re-advance could be considered. Now 6 different facies are presented, and these don't really give an understanding how the cores are subdivided into units and what it means in terms of paleoenvironmental changes. Figure 7 summarize the lithological data, but it is too small to see the details and in includes to much text. The interpretations should be omitted – they are already described in the main text.**

AC: The 6 facies reflect the fact that we have three cores from different settings: the innermost Bessel fjord, the middle part of Bessel fjord (next to an outlet glacier from a nearby ice cap), and the Dove Bugt embayment. Therefore, we prefer to keep most of the facies. The exception is Facies 5 (Sand with Normal Graded Bedding), which can be merged with Facies 3a (Laminated Sand) for simplification. But we agree, Figure 7 will be revised to better present the sedimentology of these cores. See also our comments to reviewer #2 on the lithofacies analysis, that will be updated with more references to other Greenland fjord/coastal studies.

AC: Facies 5 has been removed and the introductory paragraph to the lithology section has been simplified. Fig. 7 has been redone. The core information is easier to see and the palaeoenvironment information has been removed.

**RC: The chronology is poorly constrained and based on 5 14C ages in 3 cores. It is therefore not appropriate to calculate an accumulation rate and particularly not in the two cores where the chronology is based on a single 14C age.**

AC: Unfortunately, the cores were in part almost fossil barren (as is detailed in our reply to Review#2). Therefore, with these cores, this accumulation rate is the best that we can do with such little data and acts as a first approximation. For this reason, we would like to keep it.

AC: These have been kept and they are in Table 5. An explanation as to how these are just first approximations can be found at lines 657-659.

RC: There are some inconsistencies in the way 14C and cosmo ages are reported. Some 14C ages are reported as ka others as cal ka BP or cal BP. Sometimes the unit is not even given (see minor comments).

AC: 14C ages will all be reported the same way when possible. We will also recalibrate 14C ages for our Figure 10 (see comment above).

AC: This has been changed to "ka cal BP" throughout the manuscript, except when date is fully written out, in which case it is written as "cal yr BP".

RC: 4) The discussion is very unfocused. The paper would benefit from a clear structure where A) the first part of the discussion summarizes the main findings in the new study. B) the second part of the discussion should elaborate how the new data ties into the existing knowledge about the glaciation history of the area. Here the comparison with paleoclimate data would be relevant.

AC: Here we will follow Reviewer #2 who does not comment on this. This means that we will keep the outline for the "Discussion" the same way as was done in previous publications from our group, including Olsen et al., 2020, 2022 – both studies on similar topics from other parts of NE Greenland.

However, the section on "Sea ice cover during the Holocene" will be removed. The rest of the text will be updated to include a comparison with other studies published in 2022, 2020 and 2007, where relevant, and in accordance with the recommendations from Reviewer #2.

Olsen, I. L., Forwick, M., Laberg, J. S., and Rydningen, T. A.: Last Glacial ice-sheet dynamics offshore NE Greenland – a case study from Store Koldewey Trough, The Cryosphere (2020).

Olsen, I. L., Laberg, J. S., Forwick, M., Rydningen, T. A., and Husum, K.: Late Weichselian and Holocene behavior of the Greenland Ice Sheet in the Kejser Franz Josef Fjord system, NE Greenland, Quat. Sci. Rev. (2022).

AC: As it was stated in the original response the structure of the discussion was kept the same although the section on "sea ice cover during the Holocene" was removed. The remainder of the Discussion was updated with new information from other publications.

RC: Minor issues:

**RC: Line 1-3: I would suggest making a title that reflects the study i.e. glaciation history of Bessel Fjord.**

AC: This will be changed. Please see the response to Reviewer #2.

AC: Changed

**RC: Line 12: what does "this" and "its" refer to?**

AC: "This" refers to the Greenland Ice Sheets response to climatic changes and "its" refers to the Greenland Ice Sheet. This can, however, be rewritten and clarified in the text.

AC: Changed

**RC: Line 13-17: It is unclear how marine cores from the shelf can be used to determin how the GrIS responded to the HTM. In most places the GrIS was smaller than present during HTM i.e. not located on the shelf. I suggest this is reformulated.**

AC: In this instance Line 14-15 says "…Northeast Greenland shelf and adjacent fjords…", therefore a reference to the HTM is stated regarding the fjords. However, because this line seems unclear it will be rewritten.

AC: Changed

**RC: Line 19: It is generally accepted that the Northeast Greenland Ice Sheet is abbreviated NEGIS – not NGIS.**

AC: This can be changed.

AC: Changed

**RC: Line 43: change "making it the current largest" too and it is currently the largest**

AC: This can also be changed.

AC: Changed

**RC: Line 47: Larsen et al2018 is not the correct citation. It is better to cite Joughin et al2001 or Khan et al 2014.**

AC: This can be changed.

AC: Changed

**RC: Line 51: Omit Larsen et al2018**

AC: This can be changed.

AC: Changed

**RC: Line 60: see major concerns on relevant papers that have relevant information about NEGIS and NE Greenland in general.**

AC: This can be addressed.

AC: Changed

**RC: Line 67-72: There are more relevant papers dealing with the terrestrial deglaciation chronology that could be used and cited.**

AC: Yes, these will be added.

AC: Changed

**RC: Line 85: Move the text concerning the data source to the figure caption.**

AC: This can be moved.

AC: Changed

**RC: Line 86: change "black" box t0 "white" box**

AC: This can be changed.

AC: We reviewed this comment more closely, Fig. 1a has a black box, where Fig. 1b and Fig. 1c have white boxes. Therefore, the line about this in the caption is correct. However, on line 109 we have noted that the dashed white line of Fig. 1c outlines Fig. 2

**RC: Line 87: change "northern East" to Northeast**

AC: This can be changed.

AC: Changed

**RC: Line 110: The gravity cores have been mixed up. I (1290) is placed in the inner part of Bessel Fjord – it should be outer coast. The others are also labelled wrongly. Move the text concerning the data source to the figure caption.**

AC: You are correct, the numbering system was mistakenly mixed up. This will be corrected.

AC: Changed.

**RC: Line 137-139: There is more up-to-date literature on the Milne Land moraines. See:**

**Levy, L.B., Kelly, M.A., Lowell, T.V., Hall, B.L., Howley, J.A., Smith, C.A., 2016. Coeval fluctuations of the Greenland ice sheet and a local glacier, central East Greenland, during late glacial and early Holocene time. Geophysical Research Letters 43, 1623-1631.**

**Larsen, N.K., Søndergaard, A.S., Levy, L.B., Strunk, A., Skov, D.S., Bjørk, A., Khan, S.A., Olsen, J., 2022. Late glacial and Holocene glaciation history of North and Northeast Greenland. Arctic, Antarctic, and Alpine Research 54, d294-313.**

AC: This literature can be reviewed during our revision.

AC: This has now been updated.

**RC: Line 142: Are the ages cal ka BP, ka BP or ka?**

AC: These are calibrated 14C ages and will be updated to be in ka BP.

AC: All ages in this study have been updated.

**RC: Line 157: The first cosmogenic isotope dating on Store Koldewey was presented by Lena Håkansson. Later more data was presented by Skov and Larsen. See major concerns.**

AC: Yes, this will be added.

AC: Changed

**RC: Line 164, 166: cal ka BP**

AC: As discussed above, a single dating system will be chosen for the manuscript and values will be corrected.

AC: Changed

**RC: Line 169: no units after age?**

AC: This can be corrected. It should be "ka BP".

AC: Changed

**RC: Line 211-213: You could consider calibrating the dates with Marine20 and the suggested dR values of Pienkowski et al2022 Boreas.**

AC: Please see the comment above.

AC: We recalibrated our dates (Table 5) as well as dates from other publications (Table 3) using Marine20. However, we used a dR value of -10 ± 60 in conformity with Jackson et al. 2022). This has been now descripted in our Material and Methods section, Line 325

**RC: Line 241: The yellow color (hill-hole pair) on Fig 4b is difficult to see.**

AC: This can be changed.

AC: Changed

**RC: Line 268: Cold-based ice is not very likely under an ice stream where MSGL are formed. It is also suggested that frozen sediments have been dislocated. However, it seems unlikely that the seabed is frozen at 400m water depth. This need to be elaborated more.**

AC: We do disagree here. There are examples from the Norwegian continental shelf showing glacitectonic features formed subglacially at several hundred meters water depth (see for instance Ottesen et al. 2005), therefore, we will keep this part.

Ottesen, D., Dowdeswell, J. A., and Rise, L.: Submarine landforms and the reconstruction of fast-flowing ice streams within a large Quaternary ice sheet: The 2500-km-long Norwegian-Svalbard margin (57°-80°N), Bull. Geol. Soc. Am., 117, 1033–1050, https://doi.org/10.1130/B25577.1, 2005.

AC: Not changed, see above comment.

**RC: Line 299: The mapped landforms are very difficult to see. Consider making the figure bigger or cut the section into two.**

AC: This can be made easier to read.

AC: The figure has been made larger and turned sideways, line 450

**RC: Line 311: Again the yellow colors are very difficult to see and fig. 6b**

AC: This can be changed.

AC: Changed

**RC: Line 356-367: Rewrite and simplify.**

AC: This can be done.

AC: Changed

**RC: Line 393-394: It is not clear how subglacial or proglacial processes could have made the microfractures observed in the core. Or did the ice override the sediments after deposition. This should be elaborated.**

AC: This is a good point. This interpretation can be reconsidered or elaborated upon.

AC: Updated

**RC: Line 468-495: The section concerning the 14C dating could be simplified. And the section concerning the sediment accumulation rate should be omitted – it is not scientific sound to make this estimation with so few 14C ages.**

AC: Please see our comment above.

AC: See out above comments

**RC: Line 483: Remove Calib from table and remove accumulation rates from table (see above).**

AC: Please see our comment above.

AC: See Table 3 and 5 concerning calibration dates. See above comments about accumulation rates.

**RC: Line 496-507: this section is not well-incorporated into the description and interpretation of the data. It could potentially be incorporated in the sediment description or omitted.**

AC: This section will be deleted.

AC: Changed

**RC: Line 514-516: This sentence seems misplaced and can be omitted.**

AC: This can be deleted.

AC: Changed

**RC: Line 526: The last sentence seems misplaced?**

AC: Yes, this is something that should have been deleted earlier.

AC: Changed

**RC: Line 530-531: Remove "acted as …..Stokes and Clark) and"**

AC: We would like to keep this.

**RC: Line 543: This paleogeographic reconstruction should be updated with the newest studies from the area. See list of papers under major concerns. Much of the text is very small (font size) and could be moved to the figure caption.**

AC: This can be made easier to read and can include updated material where it is appropriate.

AC: This has been updated

**RC: Line 549-550: Nanok II is not mentioned in Vasskog et al, 2015 – they only discussed the Milne Land moraines. Probably the Nanok moraines correlated to the Milne Land moraines and they date to early Allerød – early Younger Dryas (See Larsen et al.,2022 for discussion).**

AC: This mistake is to be corrected.

AC: Changed

**RC: Line 610-611: "east of Dove Bugt in Store Koldewey.." Rewrite sentence**

AC: This can be rewritten.

AC: Changed

**RC: Line 615: I don't understand why the radiocarbon age from 1309 is a maximum age for the onset of deglaciation. Both the 1309 site and Bessel Fjord could have been deglaciated earlier. There is more than 50 cm of undated sediments below the dated interval in 1309 i.e. the site was deglaciated for some unknown time.**

AC: This should be a minimum age – to be corrected.

AC: Changed

**RC: Line 645-649: This is very speculative and not supported by any data from the study. Should be omitted.**

AC: This can be deleted.

AC: Deleted

**RC: Line 653-658: It is not scientific sound to base your interpretation on a linear age-depth model constrained by one date.**

AC: We would like to keep this as is because it is the best estimate possible with the data that is available.

**RC: Line 657: change believed with demonstrated.**

AC: This can be rewritten.

AC: Changed

**RC: Line 665-666: There are no exposure ages that show that the HTM ended by 5.6-4 ka from NE Greenland. And Briner does not provide any 10Be ages in the review paper from 2016.**

AC: You are correct in that these dates do not come from exposure dates. This is a mistake.

However, the mentioned dates are given in Briner et al. (2016), although they are a combination of results from two different regions near our study area. It was a combination of the end dates from the "summer temperature higher than present" on Hochstetter Forland based on pollen assemblages (5.6 ka) and the end of the "warmest millennia" (4 ka) from chironomids in lakes on Store Koldewey. In our revised manuscript these regions/dates will be presented separately and will include their original references. As they are technically dates for different things, they should not have been combined here.

AC: Changed

**RC: Line 673-688: This section is very speculative and the data to support presence or absence of sea ice in Bessel Fjord needs much more documentation. It is not clear from the result section that the Ca/Fe content is interpreted as a sea ice proxy.**

AC: This will be removed from the manuscript and has also been mentioned in the response to reviewer #2.

AC: Changed

**Reviewer #2 Comments and Responses:**

AC: Thank you for taking the time to carefully read our manuscript and write useful comments. You have made some very good points that we will address during our revision. We are confident that your input will ultimately improve the quality of our manuscript. Please see our comments below. We have addressed each comment one by one, with the original comments in bold and labelled "RC" and our response labelled "AC".

RC: Title: The title could be simplified (do not need the 2nd 'NE Greenland') and also should be re-worded to reflect the content more accurately. Ought to emphasise the region of study more clearly – Dove Bugt… While this does receive the outflow from Storstrømmen (southern part of NEGIS system) it also receives ice flow from L. Bistrup Bræ.

AC: Yes, here we agree, the title will be refocused to better reflect the data presented in the manuscript. Our new title is: A high-Arctic fjord system from the Last Glacial Maximum to the Present: Bessel Fjord and SW Dove Bugt, NE Greenland.

AC: The new title is changed to: "A high-Arctic inner shelf–fjord system from the Last Glacial Maximum to the Present: Bessel Fjord and SW Dove Bugt, NE Greenland"

RC: Minor point – in the literature the Northeast Greenland Ice Sheet is more commonly abbreviated to NEGIS rather than NGIS.

AC: OK, this will be changed.

AC: Changed throughout

RC: This is an interesting paper that presents important new data from an area with limited previous information from the marine realm. The paper presents swath bathymetry data to identify the signature of ice stream flow/recession in the region. This data nicely shows the direction of ice flow and can be used to help in the interpretation of ice dynamics in this region. There are also a series of sediment cores for part of the region (Bessel Fjord and Store Bælt) that provide more detailed information on sediment stratigraphy and, to a certain extent, provide a chronology for ice dynamics. This provides a minimum constraint on deglaciation of the region and also some information on timing of ice retreat within Bessel Fjord (though the chronology is rather limited, only 5 radiocarbon dates from 3 cores).

While the link with the major NGIS (or NEGIS) system is important, I would suggest increasing the emphasis on the actual region of the study, which is to the south of the main NGIS system.

AC: Yes, this will be better clarified in the "Introduction" and "Regional Setting and Environmental History" and the remaining parts of the manuscript.

AC: Less of an emphasis was made on the NEGIS and more of an emphasis was made on the study area throughout the manuscript.

**RC: Introduction**

There is a strong focus on NGIS, ok as this is an important ice stream, but need to also cover the region to the south and the other key glaciers that flow into Dove Bugt (ie L. Bistrup Bræ, and others?). Comment is made on relatively sparce research investigating NGIS mentioning some relevant papers, however, in the last few years there have been significant additional papers focussing on paleoceanography and deglaciation of NGIS that are not cited here. Examples:

Davies, J., Møller Mathiasen, A., Kristiansen, K., Hansen, K.E., Wacker, L., Olsen Alstrup, A.K., Munk, A.L., Pearce, C., Seidenkrantz, M-S., 2022. Linkages between ocean circulation and the Northeast Greenland Ice Stream in the Early Holocene. Quaternary Science Reviews 286, 107530, https://doi.org/10.1016/j.quascirev.2022.107530.

Hansen, K.E., Lorenzen, J., Davies, J., Wacker, L., Pearce, C., Seidenkrantz, M-S., 2022. Deglacial to Mid Holocene environmental conditions on the northeastern Greenland shelf, western Fram Strait. Quaternary Science Reviews 293, 107704. https://doi.org/10.1016/j.quascirev.2022.107704.

Pados-Dibattista, T., Pearce, C., Detlef, H., Bendtsen, J., Seidenkrantz, M.S., 2022. Holocene palaeoceanography of the Northeast Greenland shelf. Climate of the Past 18, 103-127.

Rasmussen, T. L., Pearce, C., Andresen, K. J., Nielsen, T. & Seidenkrantz, M.-S., 2022. Northeast Greenland:ice-free shelf edge at 79.4°N around the Last Glacial Maximum 25.5–17.5 ka. Boreas 51, 759–775. https://doi.org/10.1111/bor.12593.

Syring, N., Lloyd, J.M., Stein, R., Fahl, K., Roberts, D.H., Callard, L., O'Cofaigh, C., 2020. Holocene interactions between glacier retreat, sea ice formation, and Atlantic water advection at the inner Northeast Greenland continental shelf. Paleoceanography Paleoclimatology 35, e2020PA004019. https://doi.org/10.1029/2020PA004019.

Zehnich, M., Spielhagen, R. F., Bauch, H. A., Forwick, M., Hass, H. C., Palme, T., Stein, R., Syring, N., 2020. Environmental variability off NE Greenland (western Fram Strait) during the past 10,600 years. The Holocene 30(12), 1752–1766. https://doi.org/10.1177/0959683620950393

AC: Regarding other glacial bodies south of Storstrømmen/the NGIS: in a modern context, L. Bistrup Bræ flows northward, west of Bessel Fjord, and meets with Storstrømmen in Dove Bugt (Rignot et al. 2022). An outlet to the glacier Soranerbræen can be found in western Bessel Fjord, however, a study by Krieger et al. (2020) indicates that modern Soranerbræen has a separate drainage basin from L. Bistrup Bræ. Based on this, L. Bistrup Bræ may have a larger impact in Dove Bugt, north of the study area, rather than Bessel Fjord. Of course, this does not account for changes in ice flow patterns over the duration of time that this study focuses on. This makes L. Bistrup Bræ's impact on Bessel Fjord in the past difficult to examine in this study. Outside of Storstrømmen/the NGIS, L. Bistrup Bræ and Soranerbræen, there are no other major glacial bodies that would be relevant to the study area (other than local ice caps).

However, given the presence of large, north-south oriented glacial lineations in southwestern Dove Bugt (i.e., Store Bælt), there was a decision to focus on the southern branch of the NGIS as it likely had a larger impact on the overall region of northeast Greenland. To your point though, as much of the data is collected within Bessel Fjord rather than Store Bælt, there can certainly be more of an emphasis placed on Soranerbræen in the manuscript.

Additional information to be included is Biette et al. (2020) evidence for a late glacial, Early Holocene and late Holocene glacial advance on mountain glaciers on Clavering Island, south of our study area.

See also our comment to Reviewer #1 on this.

References here are:

Biette, M., Jomelli, V., Chenet, M., Braucher, R., Rinterknecht, V., and Lane, T.: Mountain glacier fluctuations during the Lateglacial and Holocene on Clavering Island (northeastern Greenland) from 10Be moraine dating, Boreas, 49, 873–885, https://doi.org/10.1111/bor.12460, 2020.

Krieger, L., Floricioiu, D., and Neckel, N.: Drainage basin delineation for outlet glaciers of Northeast Greenland based on Sentinel-1 ice velocities and TanDEM-X elevations, Remote Sens. Environ., 237, 111483, https://doi.org/10.1016/j.rse.2019.111483, 2020.

Rignot, E., Bjork, A., Chauche, N., and Klaucke, I.: Storstrømmen and L. Bistrup Bræ, North Greenland, Protected From Warm Atlantic Ocean Waters, Geophys. Res. Lett., 49, https://doi.org/10.1029/2021GL097320, 2022.

AC: Additional information regarding other important glaciers, such as L. Bistrup Bræ (e.g., 121-135) and Soranerbræen, has been added.

Recommended literature has been added as well:

Davies et al., 2022 lines 70, 87, 198, 818

Hansen et al., 2022 line 70,

Pados-Dibattista et al., 2022 line 198

Rasmussen et al., 2022 line 76

Syring et al., 2020 line 69

Zehnich et al., 2020 was not included

We have added a number of additional publications as well, including but not limited to:

Biette et al., 2020 line 252

Krieger et al., 2020 line 125

Rignot et al., 2022: line 123, 135, 896, 900

**RC: I think the introduction ought to put more emphasis on the actual area of study – Dove Bugt, Store Bælt and Bessel Fjord rather than NGIS, which is actually significantly further north (even though Storstrommen most likely flowed through Dove Bugt at the LGM).**

AC: We will provide more details on the modern glaciology of the study area (see comment above).

AC: This has been changed.

**RC: Regional setting**

**This section seems to suggest that L. Bistrup Bræ is also part of the NGIS system – I didn't think it was? This is an important point as there seems to be a lot of focus on NGIS, but Dove Bugt also receives ice flux from glaciers further south than NGIS I think.**

AC: This can be rephrased in the manuscript so that it is clearer to the reader that at present L. Bistrup Bræ flows northwards towards the modern southern outlet of the NGIS. Also please see my above comment regarding this.

AC: Changed

**RC: There is a brief mention of the East Greenland Current, but it would be useful to provide a little more detail on the oceanographic context of the region – is the region dominated by the cold waters of the East Greenland Current or does Atlantic Water get into Dove Bugt at depth? A recent paper by Rignot et al suggests Atlantic Water doesn't get into Dove Bugt – would be useful to cite this publication.**

**Rignot, E., Bjork, A., Chauche, N., & Klaucke, I. (2022). Storstrømmen and L. Bistrup Bræ, North Greenland, protected from warm Atlantic Ocean waters. Geophysical Research Letters, 49, e2021GL097320. https://doi.org/10.1029/2021GL097320**

AC: This is an important point, and this topic and publication will be incorporated into the manuscript.

AC: Please see lines 131-135 regarding Rignot et al. 2022 and please see lines 183-204 regarding additional information related to oceanography.

**RC: In the discussion of the wider context of glaciation of the shelf – evidence of MSGL, grounding zone wedges and ice reaching the shelf break – it would be useful to be more specific about where the evidence is from. I think most of the evidence is from further north? Are there studies from the outer sections of Store Koldewey Trough and Dove Bugt Trough?**

AC: Specific geographic location of these landforms can be added into the revised manuscript. Landforms have been identified east of the study area in Store Koldewey Trough by Laberg et al. (2017) and later with a more expanded dataset by Olsen et al. (2020).

Laberg, J. S., Forwick, M., and Husum, K.: New geophysical evidence for a revised maximum position of part of the NE sector of the Greenland ice sheet during the last glacial maximum, Arktos, 3, https://doi.org/10.1007/s41063-017-0029-4, 2017.

Olsen, I. L., Forwick, M., Laberg, J. S., and Rydningen, T. A.: Last Glacial ice-sheet dynamics offshore NE Greenland – a case study from Store Koldewey Trough, The Cryosphere Discussions, 2020.

AC: This has been added 147-149.

**RC: Overview of climate during the Holocene is provided based on regional terrestrial studies, given the references made to studies from marine region it would be useful to add a brief overview of the marine conditions during the Holocene and the estimate of HTM (this is reviewed in some of the recent papers listed above).**

AC: An "overview of the late glacial – Holocene marine conditions" section will be added to the manuscript. Additionally, mentions of Holocene conditions based on recent findings from, for example,

core PS100/270 collected east of 79°N Glacier in Syring et al. (2020), core 92G in Norske Trough in Davies et al. (2022) will be added. This will also be brought up again in the Discussion.

Davies, J., Mathiasen, A. M., Kristiansen, K., Hansen, K. E., Wacker, L., Alstrup, A. K. O., Munk, O. L., Pearce, C., and Seidenkrantz, M. S.: Linkages between ocean circulation and the Northeast Greenland Ice Stream in the Early Holocene, Quat. Sci. Rev., 286, 107530, https://doi.org/10.1016/j.quascirev.2022.107530, 2022.

Syring, N., Lloyd, J. M., Stein, R., Fahl, K., Roberts, D. H., Callard, L., and O'Cofaigh, C.: Holocene interactions between glacier retreat, sea ice formation, and Atlantic water advection at the inner Northeast Greenland continental shelf, Paleoceanogr. Paleoclimatology, 35, 2020.

AC: Information regarding this topic can be found between 183-204 and 257-276.

**RC: Methods**

**Radiocarbon dating of forams – the authors ought to outline why they are using Marine13 calibration curve rather than the updated Marine20. There are legitimate reasons why, but ought to be briefly justified here. All of the recent papers listed above actually use Marine20 – Hansen et al has a good discussion of the choice of calibration curve to use. It actually makes little difference which curve you use (though you would need to change the delta R value if using Marine20) – and both are problematic in these high latitudes, but my preference would be to use Marine20 as this is now the widely used calibration curve in this area. Also make sure there is a clear justification of the delta R value used.**

**Perhaps more importantly, it would be useful to provide a clearer rationale for the choice of position for radiocarbon dating in each core. This doesn't seem to be very clearly justified in the text.**

AC: Positions were strategically selected in each core (e.g., near unit boundaries). Additional samples were also collected within the cores, however, there was not enough datable material recovered to produce radiocarbon dates as parts of the cores were unfortunately almost carbon fossil barren. The reason for this is presently not known, although Olsen et al. (2020) also encountered this issue when analyzing cores from the nearby shelf trough, the Store Koldewey Trough. This will be stated in the "Materials and Methods" section.

AC: Line 313 – discussion on how we sampled.

AC: We have updated our radiocarbon dates (Table 5) and those from other publications (Table 3) to be recalibrated using Marine20. See more about this in our Methods section.

**RC: Results**

**Figure 2: check the labelling of the sediment cores, I think it is the wrong way round (compared to other figures).**

AC: You are correct, this needs to be changed.

AC: Changed

**RC: Figure 4: panel 4b refers to Fig 3c, 3d and 3e when it should refer to fig 4c, d and e.**

AC: This must also be changed.

AC: Changed

**RC: Figure 5: it is difficult to see the details of the annotations in 5b, though they are clearer in the enlarged sections shown in fig 6.**

AC: This can be changed to try to make it easier for the reader to see.

AC: Fig. 5 has been made larger and easier to see line 450

**RC: Seafloor landforms: There is some really good data presented here, seems to be well described and interpreted. Perhaps could draw on other studies from the Greenland margin to help support some of the interpretation.**

AC: Yes, this can be done.  Some examples of studies to draw upon:

Batchelor, C. L., Dowdeswell, J. A., and Rignot, E.: Submarine landforms reveal varying rates and styles of deglaciation in North-West Greenland fjords, Mar. Geol., 402, 60–80, 2018.

Evans, J., Ó Cofaigh, C., Dowdeswell, J. A., and Wadhams, P.: Marine geophysical evidence for former expansion and flow of the Greenland Ice Sheet across the north-east Greenland continental shelf, J. Quat. Sci., 24, 279–293, https://doi.org/10.1002/jqs, 2009.

Jakobsson, M., Hogan, K. A., Mayer, L. A., Mix, A., Jennings, A., Stoner, J., Eriksson, B., Jerram, K., Mohammad, R., Pearce, C., Reilly, B., and Stranne, C.: The Holocene retreat dynamics and stability of Petermann Glacier in northwest Greenland, Nat. Commun., 9, https://doi.org/10.1038/s41467-018-04573-2, 2018.

Newton, A. M. W., Knutz, P. C., Huuse, M., Gannon, P., Brocklehurst, S. H., Clausen, O. R., and Gong, Y.: Ice stream reorganization and glacial retreat on the northwest Greenland shelf, Geophys. Res. Lett., 44, 7826–7835, https://doi.org/10.1002/2017GL073690, 2017.

Olsen, I. L., Laberg, J. S., Forwick, M., Rydningen, T. A., and Husum, K.: Late Weichselian and Holocene behavior of the Greenland Ice Sheet in the Kejser Franz Josef Fjord system, NE Greenland, Quat. Sci. Rev., 284, 107504, https://doi.org/10.1016/j.quascirev.2022.107504, 2022.

Slabon, P., Dorschel, B., Jokat, W., Myklebust, R., Hebbeln, D., and Gebhardt, C.: Greenland ice sheet retreat history in the northeast Baffin Bay based on high-resolution bathymetry, Quat. Sci. Rev., 154, 182–198, https://doi.org/10.1016/j.quascirev.2016.10.022, 2016.

This has been updated throughout.

**RC: Sediment cores: It is actually very difficult to see the details in fig. 7 – this is an important figure to support the discussion. This figure needs to be much clearer - would be better if these logs were larger to allow reader to see the important details. Have identified a series of facies based on data presented, then provide an interpretation. Figure 8 provides examples of the facies – I think it would be helpful if the facies were actually labelled on/beside the core images in fig 8 and more clearly in Fig 7 – the text in fig 7 is not legible and makes it difficult to really see how well the core material supports the interpretation/discussion. This would make it easier to see examples of the 6 facies that have been identified – a really important part of the interpretation presented in the paper.**

AC: Figure 7 will be recreated to make it easier to see and read.

AC: Fig. 7 has been recreated.

AC: Facies are now labelled on Fig. 8.

**RC: Interpretations of facies – reference is made to supporting literature for interpretations, but I think it would be helpful to use examples from a Greenland context more here (for example in section 4.3.1. – refer to studies that have identified laminated sequences elsewhere around Greenland – see below). There are many studies that describe typically facies from ice proximal through to ice distal and also sub-ice shelf from the Greenland margin that would be useful to include here (e.g. recent studies from Ryder Glacier, O'Regan et al., 2021, The Cryosphere, 15, 4073–4097, 2021. https://doi.org/10.5194/tc-15-4073-2021 and Petermann Glacier, Reilly et al. 2019. Holocene break-up and reestablishment of the Petermann Ice Tongue, northwest Greenland. Quaternary Science Reviews218:322–42. doi:10.1016/j.quascirev.2019.06.023). Studies often use additional microfossil evidence to support proximal vs distal glaciomarine interpretations – whilst this isn't included in the current paper, reference to other recent studies from the Greenland margin that use such additional data to support facies interpretation would be beneficial here.**

AC: This is a very good suggestion. We can certainly look though O'Regan et al (2021), Reilly et al. (2019) and other Greenland publications to look for similar facies in Greenland marine cores.

AC: Examples from Greenland have been added.

**RC: Facies 1 – laminated mud – interpretation as proximal glaciomarine… ok, but would be good to use more examples from Greenland margin to help support this interpretation. There are also suggestions in the literature that such laminated sequences could be related to the presence of an ice shelf. Do you think this is an option here? An important question to consider would be how you could differentiate between an ice proximal vs ice distal interpretation for these deposits. Variation in relative sedimentation rate might help (see comments below)?**

AC: It is difficult to definitively say whether there is an ice shelf here, however ice shelves have not been identified in nearby fjords with sea-terminating outlet glaciers (e.g., Waltershausen Gletscher), indicating that this may not have been the case also for Dove Bugt – Bessel fjord during the deglaciation.

Literature from Greenland will be used to support our interpretations. Below is an example:

Evans et al. (2002) found laminated muds in cores from Kejser Franz Joseph Fjord and Fosters Bugt and believes that this indicates deposition from turbid meltwater plumes in an ice-proximal environment. They also found massive muds, like our Facies 2 and suggest that this may indicate meltwater from ice- or fjord margin-distal conditions.

Evans, J., Dowdeswell, J. A., Grobe, H., Niessen, F., Stein, R., Hubberten, H. W., and Whittington, R. J.: Late Quaternary sedimentation in Kejser Franz Joseph Fjord and the continental margin of East Greenland, Geol. Soc. Spec. Publ., 203, 149–179, https://doi.org/10.1144/GSL.SP.2002.203.01.09, 2002.

AC: This has been updated with references to Greenland and a comment about ice proximal vs distal.

**RC: Facies 2 – massive mud – the interpretation here is ice distal, however, it is not immediately clear what the basis of this interpretation is. How would you differentiate between ice distal and ice proximal environments? This is where biostratigraphy can be helpful. It may be correct, but I think it would be useful to have a stronger justification. Again, this could be with reference to other studies around Greenland that make similar interpretations (often supported by biostrat data).**

AC: Our personal thoughts on this are that in an Arctic fjord setting grain size variability is likely the result of meltwater (seasonal variability). Therefore, massive mud would likely not be found in an ice proximal environment, rather, laminated sediments. Laminated sediments can also be found in tidal environments; however, this is less likely due to the need for a high sedimentation rate.

Reference to other studies in Greenland can be used to support this proximal vs distal environment interpretation. An example of massive mud being associated with an ice distal environment is given above in our response to Facies 1. The scarcity of biological material from these cores does make biostratigraphy challenging.

AC: This has also been updated with references to Greenland and a comment about ice proximal vs distal.

**RC: Facies 3a – ok, but perhaps provide reference to published examples of such an interpretation.**

AC: Agreed, this will be added. For example, Smith & Andrew (2000) has found massive sand beds that are interpreted as sediment gravity flows (i.e., debris flows or turbidity currents) in cores from East Greenland.

Smith, L. M. and Andrews, J. T.: Sediment characteristics in iceberg dominated fjords, Kangerlussauq region, East Greenland, Sediment. Geol., 130, 11–25, 2000.

AC: This has also been updated with references to Greenland.

**RC: Section 4.3.2. Chronology**

**I think it would be useful to outline a clearer rationale for dating – larger shells – I guess this is simply driven by where the shells could be found. However, were specific depths identified for sampling of foraminifera and what was the rationale for selecting these depths? Was it linked to trying to date transitions between facies/sedimentary units?**

AC: Precisely. There was not an abundance of datable material, therefore, large shells were dated. I did touch on this above but, yes, specific positions were strategically selected. For example, one sampled was taken near the base of the core in Dove Bugt at the transition two distinctly different sediment types. And from this we were able to establish a "minimum" age of deglaciation. Another was sampled in a Bessel Fjord core at the onset of turbidites. This was interpreted as an increase in sediment input from nearby ice caps.

AC: Information about this is included in our Methods section (also see above).

**RC: Foraminiferal species mentioned in this section – need to be careful to use capitals for genus name – e.g. Islandiella norcrossi, Stainforthia feylingi (lines 477, 478). Were samples taken systematically through the cores to try to improve the chronology – this seems to be a major weakness of this study – the rather limited number of radiocarbon dates from the cores.**

AC: Please see the above comment. The strategy was to date immediately above or below lithofacies boundaries, but a lack of datable material left us with the samples dated and presented in this study. This was also an issue for Olsen et al., (2020) when analyzing cores from Store Koldeway trough and from the Kejser Franz Josef fjord system, where we also ended up with just a few levels with enough material for dating (Olsen et al., 2022) – here an even higher number of samples were checked.

AC: Corrected

**RC: Discussion of sedimentation rates – ought to acknowledge that the core surface is unlikely to be the present day due to the gravity coring process. Without dating of the surface it is hard to estimate**

how much material is likely to have been lost, but ought to at least acknowledge this problem. Variation in sedimentation rates – this is likely to be strongly influenced by the proximity of the grounded ice front and the shift between proximal glaciomarine and distal glaciomarine conditions. Indeed, using the variation in sedimentation rates could be an additional way to support the interpretation of facies provided earlier in the paper.

AC: There was not a very soft-core top and the gravity corer did not over-penetrate during coring. X-rays show relatively minor disturbance. Box cores were also taken in the same positions (not presented here), and from this, as a first approximation, gravity core tops are considered to be present days sediment.

The proximity of a grounded ice front would certainly influence the sedimentation rate (proximal vs distal). In our view, this may be difficult to assess in our study as so few biological samples were recovered, making the sedimentation rate an average covering large periods of time. This point will be considered in our revision.

AC: Information regarding the core surface has been added (lines 655-657).

**RC: Discussion**

**Line 526 – there is half a sentence… need to edit.**

AC: This will be deleted.

AC: Deleted

**RC: The discussion of the bathymetric data and morphology seems reasonable.**

**Section 5.2. One of the key additional information from this paper is the potential dating of ice retreat through Store Bælt from the sediment core beyond the mouth of Bessel Fjord. It is actually difficult to see the evidence from fig 7 because the core is so small and the text too small to read. I think the cores in fig. 7 need to be increased in size so that the detail is legible. A key point is the transition from proximal to distal glaciomarine mud – this is where clearer and better supported interpretation of the facies would be useful. Is there a clear change in sedimentation rate between these sections? There is no diamicton at the base, so have not penetrated a subglacial till. Therefore the deglaciation age of 11.2 ka is a minimum age, the site must have been deglaciated before this date. Also I got confused – fig. 2 appears to suggest core 1309 is near the head of Bessel Fjord and 1290 is in Dove Bugt. Need to make sure this is corrected.**

AC: Figure 7 will certainly be made easier to read and larger.

Figure 2 does contain a mistake. The core labels I and III should be switched. This will be changed to match the numbering system from figure 5 to create consistency between the figures.

AC: Fig. 7 was completely redone and is now easier to read.

AC: Fig. 2 is also corrected.

**RC: Discussion of comparison with other estimates of deglaciation, must be clear that the date presented here is a minimum – it must have been deglaciated by 11.2 and potentially before that.**

AC: This can be done.

AC: This has been made clear in the text (ex. 760, 770, 839-849).

**RC: Deglaciation of Bessel Fjord – this is rather limited by the poor chronology from the cores recovered in the fjord. Were these cores sampled systematically to try to find calcareous benthic foraminifera for radiocarbon dating? It is a shame there aren't more dates from these cores.**

AC: Please see the above responds to this. But yes, more radiocarbon dates would have been ideal.

AC: Please see our Methods chapter

**RC: This seems to be a significant weakness of the study – the rather limited number of radiocarbon dates available to provide a chronology. More dates would also help in refining estimates of sedimentation rate that, in turn, could help to support the interpretation of relative proximity to grounded ice margin (ie ice proximal vs ice distal). However, this is often an issue in such high latitude studies, but it would be worth expanding a little in the text on how systematically samples were taken to try to improve the chronology.**

AC: Please see the comments above. An explanation about this can be added to the manuscript.

AC: Please see our Methods chapter

**RC: Discussion of sea ice cover – this is rather unconvincing. Why not use organic carbon measurements to estimate productivity? I would suggest removing this section.**

AC: This section of the manuscript will be removed and has also been mentioned in the response to reviewer #1.

AC: This was removed from the manuscript.

---

## Referee Report (RR1)

[referee-annotated manuscript omitted]

---

## Author Response (AR2)

[revised manuscript text omitted]